# Research

neuroscience

defensive states, threat, reward, fear, gaze, freezing

**Author for correspondence:**
Alma-Sophia Merscher
e-mail: alma-sophia.merscher@
uni-wuerzburg.de

# Centralized gaze as an adaptive component of defensive states in humans

Alma-Sophia Merscher[1], Philip Tovote[2], Paul Pauli[1] and Matthias Gamer[1]

[1]Department of Psychology, University of Würzburg, Marcusstr. 9-11, 97070 Würzburg, Germany
[2]Systems Neurobiology, Institute of Clinical Neurobiology, University Hospital Würzburg, Versbacher Str. 5, 97078 Würzburg, Germany

 A-SM, 0000-0003-0051-2797; MG, 0000-0002-9676-9038

Adequate defensive responding is crucial for mental health but scientifically not well understood. Specifically, it seems difficult to dissociate defense and approach states based on autonomic response patterns. We thus explored the robustness and threat-specificity of recently described oculomotor dynamics upon threat in anticipation of either threatening or rewarding stimuli in humans. While visually exploring naturalistic images, participants (50 per experiment) expected an inevitable, no, or avoidable shock (Experiment 1) or a guaranteed, no, or achievable reward (Experiment 2) that could be averted or gained by a quick behavioural response. We observed reduced heart rate (bradycardia), increased skin conductance, pupil dilation and globally centralized gaze when shocks were inevitable but, more pronouncedly, when they were avoidable. Reward trials were not associated with globally narrowed visual exploration, but autonomic responses resembled characteristics of the threat condition. While bradycardia and concomitant sympathetic activation reflect not only threat-related but also action-preparatory states independent of valence, global centralization of gaze seems a robust phenomenon during the anticipation of avoidable threat. Thus, instead of relying on single readouts, translational research in animals and humans should consider the multi-dimensionality of states in aversive and rewarding contexts, especially when investigating ambivalent, conflicting situations.

## 1. Introduction

Various forms of defensive behaviours have evolved to protect an organism from potential harm in threatening situations [1,2]. Depending on the context (e.g. availability of escape routes; temporal and spatial distance of a threat), they occur in a cascade-like fashion ranging from hard-wired automatic, initial reactions to deliberative goal-directed behaviors [3–7].

An evolutionarily conserved response in the face of real or perceived threat, which has been extensively investigated in rodents, is a defensive behavioural pattern eventually termed freezing [8]. It is characterized by movement cessation, accompanied by a transient decrease in heart rate, i.e. bradycardia [9–12]. This defensive mode of simultaneous behavioural and cardiovascular inhibition has been suggested to help in avoiding predator detection [13] optimizing perceptual and attentional processing [14,15], and to prepare fast responses to approaching threat [16,17].

Upon distal yet inevitable threat, humans seem to engage in similar defensive responding denoted by reduced body sway (i.e. freezing), as measured by stabilometric platforms, and a co-activation of sympathetic (e.g. heightened skin conductance) and parasympathetic (e.g. bradycardia) branches of the autonomic nervous system [7,18,19]. This integrated defense state has been referred to as attentive immobility/freezing, supposedly preparing the individual for further defensive actions if eventually the threat becomes imminent or escape options appear [20–22]. Confusingly, attentive immobility in humans has been discussed as both a vulnerability factor for psychopathologies

(e.g. [23–25]) and an adaptive action-preparatory mechanism [18,19,26]. To reconcile these divergent results, it has been suggested that freezing tends to be associated with clinical constructs when participants cannot escape the threat, while it adaptively facilitates action preparation when subsequent harm can be actively avoided [19]. Experimental context thus seems to constitute a main determinant of the behavioural defense state [27,28]. While attentive immobility describes a state of heightened vigilance and action preparation when escape might still be an option, a lack of escape routes upon imminent threat elicits a more desperate defense state that has also been termed 'immobility under attack' [21].

However, this explanation does not hold for all studies in this domain. For example, reduced body sway was found to be related to both vulnerability markers of psychopathologies and faster threat reactions when participants were able to actively avoid aversive stimulation. Heart rate, in contrast, was only associated with faster motor responses in this study [26]. Diverging from the widespread idea that due to its robust co-occurrence [7], bradycardia might be adduced as a proxy for freezing (e.g. [29,30]), cardiac and motor inhibition seem to reflect different aspects of a defense state. Previous studies using active contexts suggested that transient bradycardia indeed also occurs independently of threat, constituting a more general action-preparatory mechanism [31–33]. This goes along with the notion that appetitive and defensive responding bear fundamental similarities in that they require the organism to anticipate and prepare for subsequent actions [32–34]. Responses that specifically index and discriminate between threat-induced, defensive and reward-related appetitive states in humans are thus necessary to understand fear-associated neural circuitries and behaviours. Oculomotor dynamics may function as such a marker: Rösler & Gamer [35] were the first to report reduced visual scanning along with bradycardia and increases in skin conductance during the expectation of an avoidable aversive electrotactile stimulation. Specifically, participants showed fewer and longer fixations that were closer to the centre of the screen when anticipating a preventable shock that they could avoid by pressing the space bar as compared to an inevitable or no shock. Corroborating the idea of an action-preparatory mechanism, such narrowed overt attention, as well as enhanced bradycardia, predicted the speed of threat escapes on a trial-wise basis. These results align with previous findings indicating a narrowed focus of attention (i.e. lower accuracy when responding to peripheral versus central stimuli in a maze) while participants had to actively flee from a predator chasing them as compared to when they were not actively haunted [36]. However, as previous research suggests that eye movements follow task-specific predictions for expected events [37], it remains unclear whether these findings relied on the specific experimental situation that required a narrow attentional focus to prepare for the upcoming flight response.

We thus conducted two experiments to investigate whether oculomotor changes upon threat can also be observed in situations requiring a wider focus of attention and whether they are threat-specific. Using adapted versions of the said paradigm by Rösler & Gamer [35], we assessed gaze dynamics and autonomic responses during the expectation of an avoidable, no, or inevitable threat (Experiment 1) or an achievable, no, or guaranteed reward (Experiment 2). In Experiment 1, we modified the task to avert the threat such that it required more distributed spatial attention. Instead of a simple button press, subjects could now avoid the aversive electrotactile stimulation by quickly reacting upon peripherally presented response prompts. In Experiment 2, we transferred the paradigm into a new context to test whether the concomitant inhibition of oculomotor behaviour and cardiac output, resembling the previously described freezing-associated bradycardic defense state, represents a threat-specific phenomenon: instead of shocks, participants could earn a guaranteed, no, or an achievable financial reward that could be won by quickly responding to a peripherally presented stimulus.

## 2. Methods

### (a) Participants
The current study was based on a similar design to Rösler & Gamer [35], in which 50 participants were examined. This allowed the detection of medium effect sizes in repeated measures analyses of variance (ANOVAs, $f = 0.25$) at an alpha level of 0.05, with a statistical power greater than 0.95 when assuming a correlation of $r = 0.50$ between factor levels [38]. We therefore decided to acquire a similar number of valid datasets for both experiments in the current study.

Subjects (Experiment 1: $N = 58$; Experiment 2: $N = 60$) signed up for the study via an online platform and thus formed a sample of convenience. All participants had normal or corrected-to-normal vision (contact lenses). In Experiment 1, seven participants were excluded due to problematic eye-tracking data (i.e. more than 30% of eye-tracking trials with baseline outliers or missing baseline position data, or a range of baseline coordinates exceeding 5° of visual angle after trial exclusion) and another one because of frequent extrasystoles in the heart rate data, resulting in a total of 50 participants (40 women, age: $M = 28.00$ years, s.d. = 9.78 years). In Experiment 2, six participants were excluded due to problematic eye-tracking data and four due to technical errors during data recording, also forming a final sample of 50 participants (40 women, age: $M = 28.13$ years, s.d. = 10.79 years). All participants provided written informed consent and were reimbursed with 10 € (Experiment 1) or 4 € plus a variable bonus according to their performance in the study (additional 4 € max; Experiment 2).

The experiments were conducted according to the declaration of Helsinki and have been approved by the ethics committee of the University of Würzburg. All datasets generated and analysed in the current study are available on the Open Science Framework at https://osf.io/whpt5/?view_only=34f40d237f234c24b91f3a06e2c81a24.

### (b) Experimental design
Based on Rösler & Gamer [35], participants were presented with a screen depicting naturalistic pictures against a grey background during which eye movements, pupil width, heart rate (HR) and skin conductance (SC) were measured. Experiment 1 included a threat context where participants could receive an individually calibrated, aversive electrotactile stimulation using a Digitimer Constant Current Stimulator DS7A (Hertfordshire, UK; see electronic supplementary material for further details), while Experiment 2 involved a reward context where participants could earn money (fixed amount of 0.10 € per trial).

For visual stimulation, we used 60 affectively neutral images depicting naturalistic scenes (768 × 576 pixels, visual angle of 24.00° × 18.11° at a viewing distance of 50 cm) from the McGill Calibrated Colour Image Database [39] presented in a random order. Although the pictures did not include particularly salient features, half of them (randomly determined within each

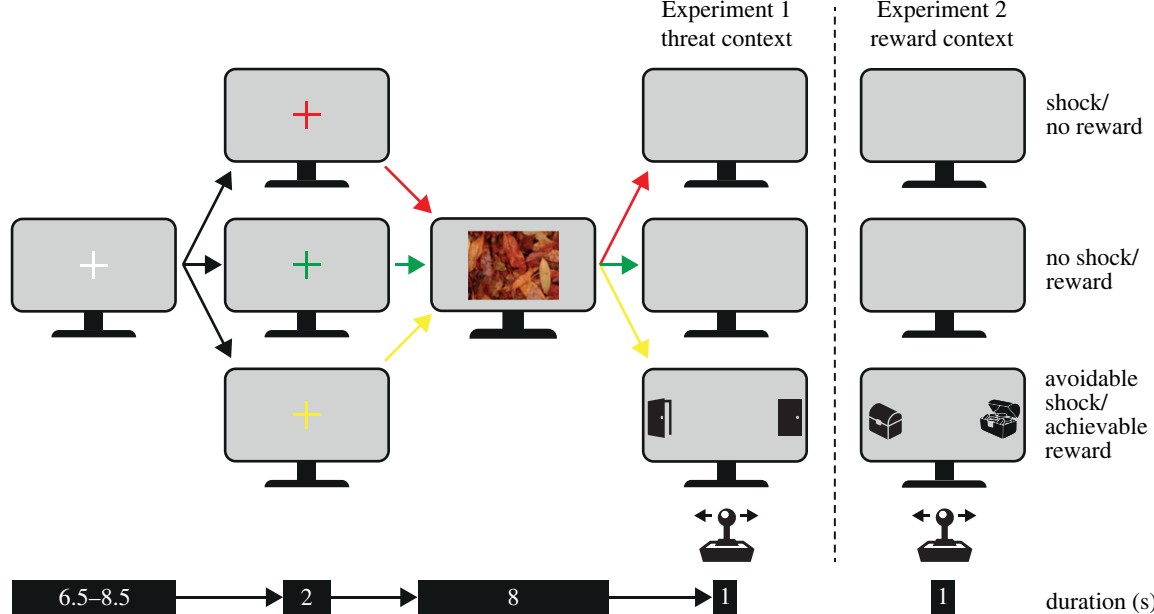

**Figure 1.** Study design of Experiment 1 (Threat context) and Experiment 2 (Reward context) adapted from Rösler and Gamer [35]. Participants were told that a coloured fixation cross would signal whether to expect an inevitable, no, or an avoidable shock (Experiment 1) or no, a guaranteed or achievable reward (Experiment 2) after an anticipation phase during which naturalistic images were presented. Avoidable shocks and achievable rewards could be averted or gained by a quick joystick movement towards an indicated side (an open door or an open treasure chest, respectively). Note that the size of colour cues and response prompts are not drawn to scale. (Online version in colour.)

participant) were horizontally flipped to prevent biases in eye movements provoked by an incidental imbalance in visually stimulating features on one side. The experiment was programmed with Presentation (Neurobehavioral Systems Inc., v. 18.1) and run on a 24″ Asus VG248QE display (53.126 × 29.889 cm, 1920 × 1080 pixels, refresh rate 60 Hz). Naturalistic images (8 s) were preceded by a white fixation cross (6.5–8.5 s) turning red, green or yellow (2 s). Depending on the colour, participants were instructed whether to expect an inevitable shock or no reward (red), no shock or guaranteed reward (green) or an avoidable shock or achievable reward (yellow) after disappearance of the naturalistic picture. Afterwards, the screen turned blank (1 s) in no shock and shock or no reward or reward trials, while the shock and reward trials were accompanied by a shock or a reward delivery. Differing from Rösler & Gamer [35] who used a verbal prompt to press the space bar as fast as possible, an open and a closed-door (Experiment 1) or treasure chest (Experiment 2) appeared on either side of the screen in avoidable shock/achievable reward trials (225 × 388 pixels, distance between the centre of the screen and the centre of the object in visual angle: 19.88°). The position of the open door or open treasure chest was counterbalanced within each participant. Participants were told they could avoid the electrotactile stimulation or gain the reward by a quick joystick movement in the direction of the open door or treasure chest (for an illustration of the experimental design, see figure 1). In order to ensure that participants would receive a shock or win a reward in approximately 50% of the trials, they had to be faster than 600 ms in the first five trials. Afterwards, the threshold was individually adjusted to the median of the first five reaction times. Both experiments comprised 60 trials with 20 in each condition (see electronic supplementary material for more information on experimental setup and procedure).

### (c) Data recording
#### (i) Eye-tracking
Movements of the right eye were measured using an EyeLink 1000Plus system (SR Research Ltd, Ottawa, Canada) in the tower mount configuration with a sampling rate of 1000 Hz. Gaze position data were parsed into fixations and saccades using EyeLink's default configuration. Saccades were defined as fast eye movements with a velocity exceeding $30°\,s^{-1}$ or an acceleration exceeding $8000°\,s^{-2}$. The last 300 ms before stimulus onset were defined as baseline and used for an offline drift correction. To ensure that participants fixated the centre of the screen during the baseline, we used our laboratory's established iterative outlier detection algorithm [19,40]. Therefore, we temporarily removed the highest and lowest values of baseline data, separately for $x$ and $y$ coordinates, from the distribution and inspected whether they deviated more than three standard deviations from the mean of the remaining data. If so, one or both of these values were labelled as outliers and permanently removed; otherwise, they were returned to the dataset. This procedure was iteratively applied to the remaining distribution until no further $x$- and $y$-values met the removal criterion. Individual gaze drift was finally corrected by subtracting the $x$- and $y$-coordinates of the baseline from the fixation coordinates during stimulus exploration. Trials with baseline outliers or missing baseline position data (Experiment 1: 8.20%, Experiment 2: 7.90%) as well as trials with premature behavioural responses during the anticipation phase (Experiment 1: 0.13%, Experiment 2: 0.07%) were excluded from all further analyses. We then computed three oculomotor metrics on a second-by-second basis: the average distance of fixations from the centre of the screen in pixels (centre bias), the duration of individual fixations, and the number of fixations. For the analyses, these three metrics were averaged into eight one-second bins covering the entire period of picture viewing.

#### (ii) Pupil width
From the eye-tracking data, we also extracted the recorded pupil diameter. In a first step, we linearly interpolated blink periods and downsampled the data to 100 Hz. Subsequently, we applied a 2 Hz low-pass filter and converted the values from arbitrary units to mm according to [41]. We then calculated changes in pupil diameter relative to a 1 s baseline period preceding cue

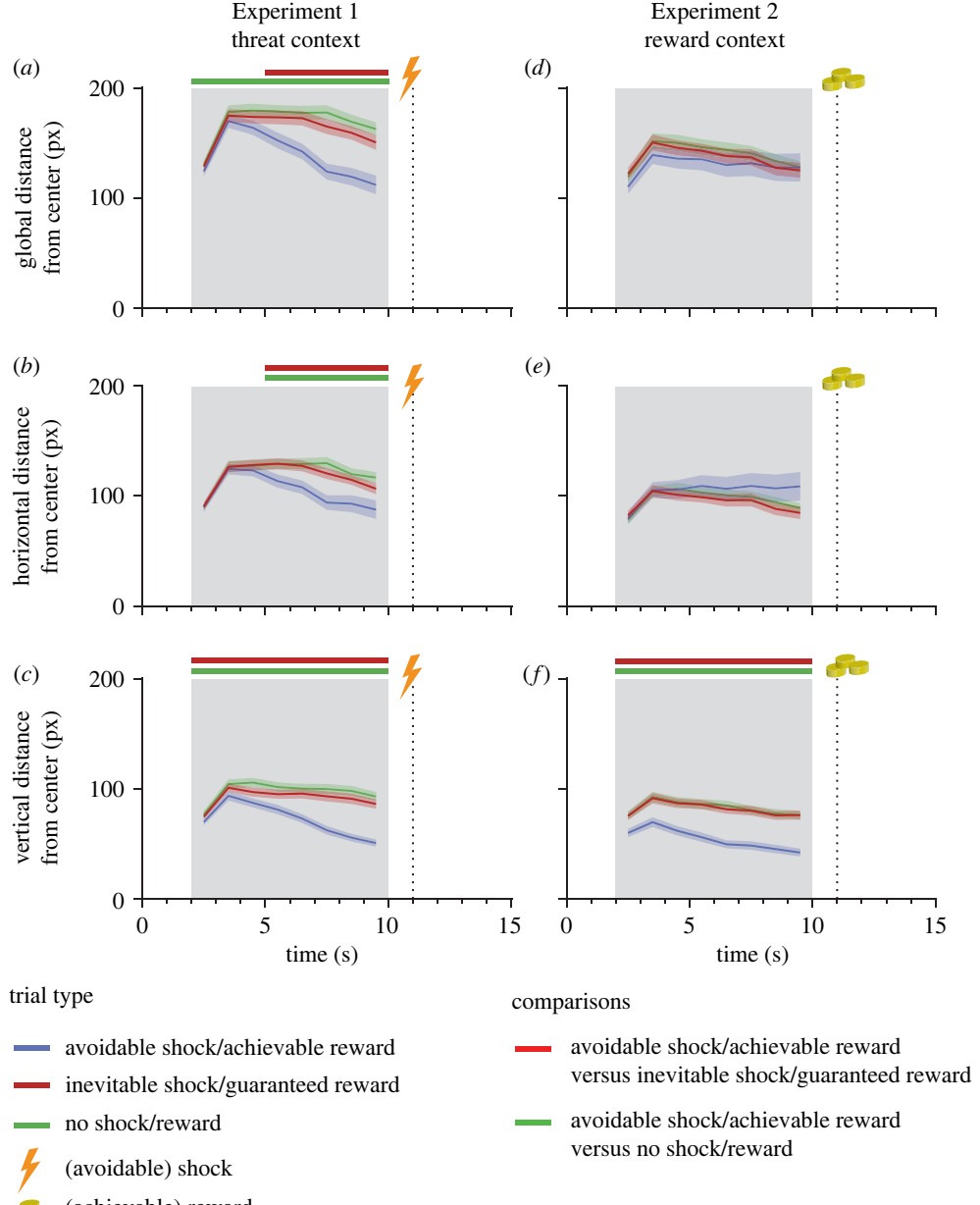

**Figure 2.** Changes in centre bias during the anticipation of an inevitable, no, or avoidable shock in Experiment 1 ((a) global centre bias; (b) horizontal centre bias; (c) vertical centre bias) or a guaranteed, no, or achievable reward in Experiment 2 ((d) global centre bias; (e) horizontal centre bias; (f) vertical centre bias). Shaded ribbons denote standard errors of the mean. Horizontal lines at the top of each figure indicate significant differences between avoidable shock (a,b and c) or achievable reward (d,e and f) trials and the other two trial types (after false discovery rate correction). Shading in grey denotes the phase between onset and offset of picture presentation, with the offset prompting quick responses in the avoidable shock and achievable reward trials, respectively. (Online version in colour.)

onset. These values were then averaged into 20 bins of 0.5 s each, spanning the whole cue and anticipation period. Note that we used a smaller bin duration for pupil size than for the other oculomotor and physiological measures since the pupil responds more quickly to external and internal events.

### (iii) Electrodermal activity

Skin conductance was recorded continuously using a BIOPAC MP160 system (BIOPAC Systems, Inc.) at a sampling rate of 500 Hz from two Ag/AgC1 electrodes filled with 0.05 ml NaCL electrolyte placed on the thenar and hypothenar eminences of the non-dominant hand. For the analyses, SC data were downsampled to 20 Hz and values were averaged into 10 one-second bins for each condition (five additional one-second bins for the post-stimulus phase were added for data visualization). The 10 s interval started two seconds prior to image onset to include the cue phase. The last second prior cue onset

served as baseline and was subtracted from all subsequent data points of each trial.

### (iv) Heart rate

An electrocardiogram (ECG) was recorded using the same BIOPAC system with disposable Ag/AgC1 electrodes placed on the right clavicle and the lower left ribcage, and with the reference electrode placed on the right lower ribcage. Sampling rate was 500 Hz. ECG data were filtered using a 2 Hz high-pass filter to remove slow signal drifts. Afterwards, R-peaks were detected semi-automatically using in-house software and manually edited in case of detection errors. R–R-intervals were converted to HR in beats per minute and a real-time scaling procedure [15] was implemented to calculate mean heart rate for 10 one-second time bins (plus 5 bins for data visualization) spanning the same time window as for SC. The HR in the last

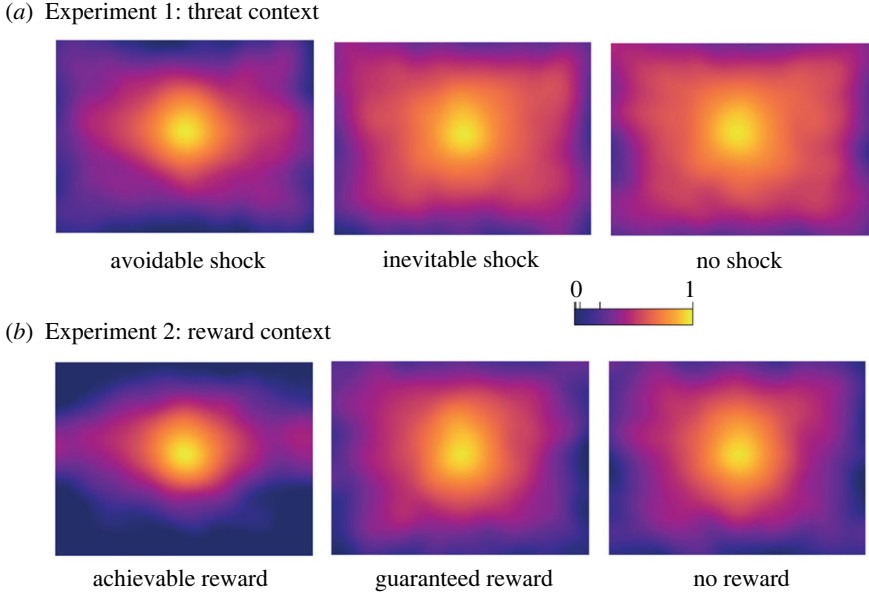

(a) Experiment 1: threat context

avoidable shock          inevitable shock          no shock

0          1

(b) Experiment 2: reward context

achievable reward        guaranteed reward        no reward

**Figure 3.** Normalized fixation density maps reflecting the distribution of fixations on the naturalistic images during the anticipation phase in (a) Experiment 1 and (b) Experiment 2. Fixation densities are depicted on a logarithmic scale. (Online version in colour.)

second prior to cue onset served as baseline and was subtracted from all following time bins.

## (d) Statistical analyses

Data preprocessing and analyses were performed using R (v. 3.3.3, R Core Team, 2018 [42]) on a significance level of 5%. For all dependent variables, we calculated repeated-measures analyses of variance (rmANOVA) with trial type (Experiment 1: inevitable shock, no shock, avoidable shock; Experiment 2: guaranteed reward, no reward, achievable reward) and second (10 or 20 bins including the cue period for skin conductance, heart rate and pupil width and 8 bins restricted to the picture presentation phase for all three metrics of visual exploration) as within-subject factors for each study. Degrees of freedom were adjusted according to Greenhouse–Geisser to compensate for potential violations of the sphericity assumption. To specifically compare the respective active condition (Experiment 1: avoidable shock; Experiment 2: achievable reward) with the remaining two conditions, *post hoc* t-tests were performed using false discovery rate correction (FDR [43]) to adjust for alpha-error accumulation (comparisons between the two passive conditions in each experiment are reported in electronic supplementary material, figures S3–S5).

We additionally computed a generalized linear mixed model (GLMM) for each study using both heart rate and centralization of gaze during the second half of the picture viewing period separately as predictors for reaction times in the avoidable shock and achievable reward trials, respectively. Subject ID was added as a random intercept into the GLMM. Further details on the calculations, the internal consistency (table S9) and correlations between autonomic and oculomotor measures (figure S2) are included in the electronic supplementary material.

## 3. Results

### (a) Oculomotor behaviour during the anticipation of threat and reward

Using a 3 × 8 rmANOVA, we first compared the average distances of fixations from the centre of the screen (global centre bias) during image presentation between inevitable, no, and avoidable shock trials. Visual exploration decreased markedly toward the end of the anticipation period when participants awaited an avertable shock as compared to both other conditions (figure 2a; interaction trial type × second, $F_{14,686} = 13.64$, $\varepsilon = 0.31$, $p < 0.001$, $\eta^2_g = 0.03$; main effects for this and the following analyses are reported in electronic supplementary material, table S1 and table S2). No robust differences were observed between the inevitable and the no shock conditions (see electronic supplementary material, figure S3). In avoidable shock trials, global centre bias predicted faster response times on a trial-wise basis ($\beta = 0.21$, $SE = 0.05$, $t_{772.73} = 3.66$, $p < 0.001$).

Contrastingly, we found differential effects in the reward-context. There were no statistically significant differences in the temporal progression of the global centre bias between reward, no reward and achievable reward trials (figure 2d; interaction trial type × second, $F_{14,686} = 0.89$, $\varepsilon = 0.19$, $p = 0.438$, $\eta^2_g < 0.01$), and global centre bias did not predict faster response times in achievable reward trials on a trial-wise basis ($\beta = -0.01$, $SE = 0.04$, $t_{720.66} = -0.27$, $p = 0.788$).

To get a more nuanced picture of the visual scanning patterns in threat and reward contexts, we generated fixation density maps. These maps show a pronounced centralization of fixations in avoidable shock trials but a more strategic gaze pattern of horizontal exploration in the reward context (figure 3). To statistically confirm this impression, we calculated distinct centre biases for the horizontal and vertical coordinates of fixations, respectively. Separate 3 × 8 rmANOVAs showed a centre bias to be evident toward the end of the anticipation phase when participants expected a potential aversive stimulation on both the horizontal axis (figure 2b, interaction trial type × second: $F_{14,686} = 6.71$, $p < 0.001$, $\eta^2_g = 0.02$) as well as the vertical axis (figure 2c, interaction trial type × second: $F_{14,686} = 19.57$, $p < 0.001$, $\eta^2_g = 0.04$). Comparable analyses for the reward context revealed an absent horizontal centre bias (figure 2e, interaction trial type × second, $F_{2,98} = 2.23$, $p = 0.097$, $\eta^2_g = 0.01$), but a strong vertical centre bias in achievable reward trials (figure 2f, interaction trial type × second, $F_{2,98} = 6.35$, $p < 0.001$, $\eta^2_g = 0.01$).

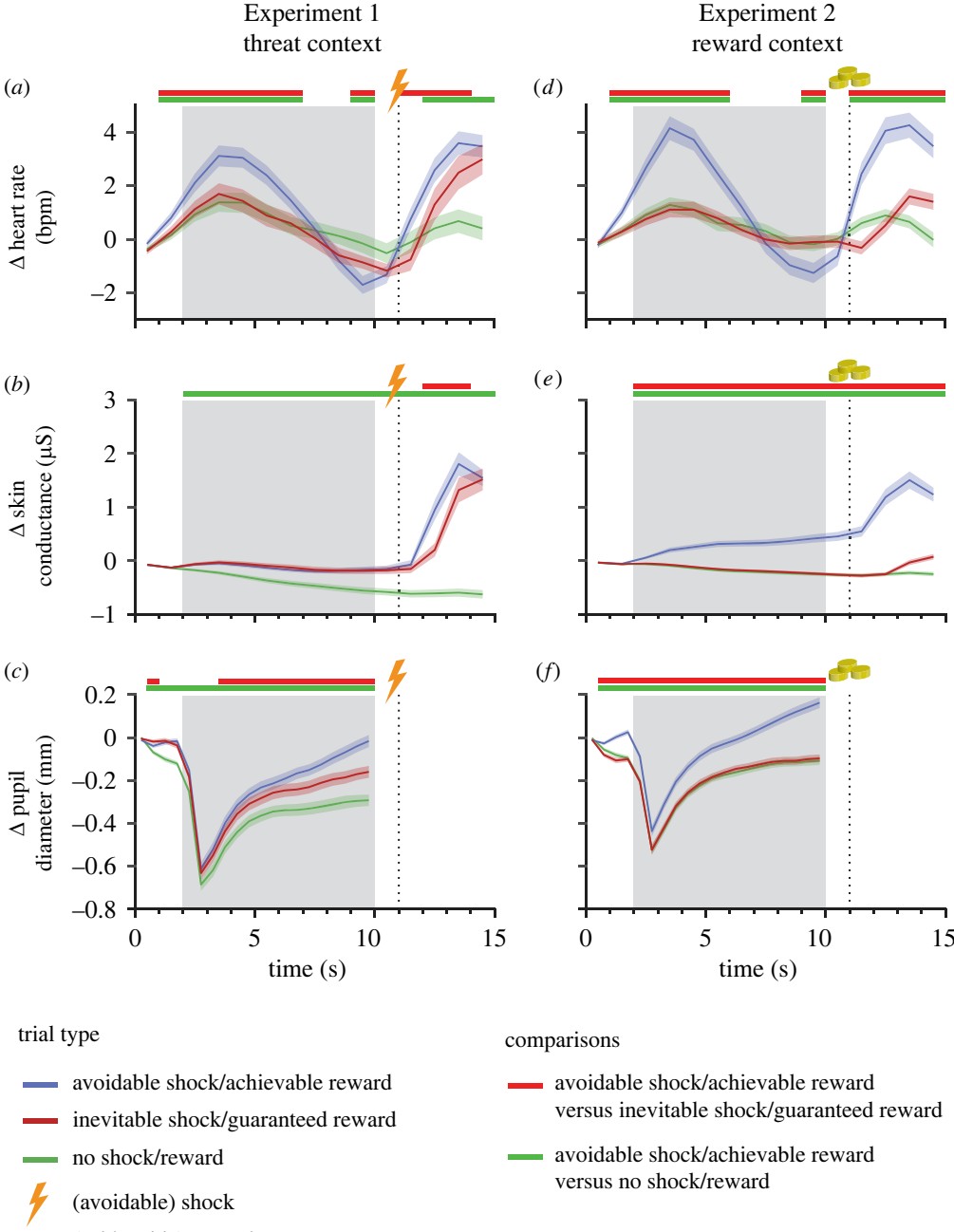

**Figure 4.** Autonomic responses during the anticipation of an inevitable, no or avoidable shock in Experiment 1 (heart rate: (*a*), skin conductance: (*b*), pupil width: (*c*)) or a guaranteed, no or achievable reward in Experiment 2 (heart rate: (*d*), skin conductance: (*e*), pupil width: (*f*)). Shaded ribbons denote standard errors of the mean. Horizontal lines at the top of each figure indicate significant differences between avoidable shock (*a,b* and *c*) or achievable reward (*d,e* and *f*) trials and the other two trial types (after false discovery rate correction). Shading in grey denotes the phase between onset and offset of picture presentation with the offset prompting quick responses in the avoidable shock and achievable reward trials, respectively. (Online version in colour.)

A direct comparison of the response patterns between both experiments using a $2 \times 3 \times 8$ ANOVA now including the between-subjects factor experiment confirmed significant differences in the observed interaction effect between experiments for general reductions in visual scanning (electronic supplementary material, table S3; three-way interaction, $F_{14,1372} = 8.31$, $\varepsilon = 0.25$, $p < 0.001$, $\eta_g^2 = 0.01$). This was also evident for both horizontal centre bias (electronic supplementary material, table S4; three-way interaction, $F_{14,1372} = 6.80$, $\varepsilon = 0.25$, $p < 0.001$, $\eta_g^2 = 0.01$) and vertical centre bias (electronic supplementary material, table S5, three-way interaction, $F_{14,1372} = 10.82$, $\varepsilon = 0.24$, $p < 0.001$, $\eta_g^2 = 0.01$). Moreover, the predictive value of globally reduced image exploration for reaction times differed significantly

between shock and reward contexts in the GLMM (interaction of experiment and amount of centre bias, $\beta = -0.22$, $SE = 0.07$, $t_{1513.10} = -3.25$, $p = 0.001$). We also compared fixation numbers and durations between trial types within each experiment but failed to find substantial effects in both experiments (see electronic supplementary material, table S1 and figure S1).

## (b) Autonomic responses during the anticipation of threat and reward

To compare heart rate changes between conditions, we performed a $3 \times 10$ rmANOVA (trial type by seconds, now also including the 2 s cue period) for each experiment. In the

aversive context (Experiment 1), average heart rate increased right after cue onset, decreased over the anticipation period and increased again after picture offset across all conditions. This dynamic became gradually more pronounced from no to inevitable to avoidable shock trials, but statistically significant differences only emerged between the avoidable shock and the other two conditions (figure 4a and electronic supplementary material, figure S4, interaction trial type × second, $F_{18,882} = 13.07$, $\varepsilon = 0.27$, $p < 0.001$, $\eta_g^2 = 0.04$). In the reward context, achievable reward trials but not guaranteed or no reward trials showed similar heart rate trends, with a marked increase after cue onset followed by a decrease and another increase after picture offset (figure 4d; interaction trial type × second, $F_{18,882} = 23.38$, $\varepsilon = 0.28$, $p < 0.001$, $\eta_g^2 = 0.09$).

Whereas heart rate deceleration during the second half of picture viewing did not significantly predict faster response times in avoidable shock trials on a trial-wise basis in a GLMM ($\beta = 0.19$, $SE = 0.58$, $t_{819.59} = 0.32$, $p = 0.745$), it did so in achievable reward trials ($\beta = 1.60$, $SE = 0.57$, $t_{803.59} = 2.81$, $p = 0.005$). However, this apparent difference between studies was not statistically significant when contrasting both experiments within one GLMM (interaction of experiment and mean heart rate, $\beta = 1.41$, $SE = 0.82$, $t_{1625.76} = 1.73$, $p = 0.085$; main effect of mean heart rate, $\beta = 0.19$, $SE = 0.56$, $t_{1611.69} = 0.34$, $p = 0.735$). A direct comparison of heart rate changes during the anticipation phase using a $2 \times 3 \times 10$ ANOVA with the additional between-subjects factor experiment revealed a significant three-way interaction, underlining that despite some similarities, heart rate changes also differed between threat and reward contexts (electronic supplementary material, table S6, $F_{18,1764} = 3.45$, $\varepsilon = 0.305$, $p = 0.003$, $\eta_g^2 = 0.01$).

Skin conductance levels increased during the anticipation of both an inevitable and an avoidable shock in comparison to no shock trials (figure 4b; interaction trial type × second, $F_{18,882} = 12.50$, $\varepsilon = 0.16$, $p < 0.001$, $\eta_g^2 = 0.04$), whereas in a reward context SC levels only increased in achievable reward trials while they remained stable in guaranteed and no reward trials (figure 4e; interaction trial type × second, $F_{18,882} = 38.68$, $\varepsilon = 0.09$, $p < 0.001$, $\eta_g^2 = 0.27$). A direct comparison between experiments using a $2 \times 3 \times 10$ ANOVA including the between-subjects factor experiment confirmed a different temporal progression of skin conductance changes as a function of trial type in both experiments (electronic supplementary material, table S7; three-way interaction, $F_{18,1764} = 17.59$, $\varepsilon = 0.14$, $p < 0.001$, $\eta_g^2 = 0.03$).

Changes in pupil size revealed a similar pattern. In the shock context, pupil width increased more strongly over time in both inevitable shock and avoidable shock trials as compared to no shock trials, with the strongest increase in avoidable shock trials (figure 4c; interaction trial type × second, $F_{38,1862} = 24.47$, $\varepsilon = 0.15$, $p < 0.001$, $\eta_g^2 = 0.03$). In the reward context, only achievable reward trials contrasted with the other two trial types were associated with a strong increase in pupil width over the course of the whole trial (figure 4f; interaction trial type × second, $F_{38,1862} = 35.63$, $\varepsilon = 0.08$, $p < 0.001$, $\eta_g^2 = 0.06$). Comparing these interaction effects between experiments confirmed a significant difference between shock and reward contexts regarding differential changes in pupil width between trial types over time (electronic supplementary material table S8; three-way interaction, $F_{38,3724} = 2.39$, $\varepsilon = 0.13$, $p = 0.039$, $\eta_g^2 < 0.01$).

# 4. Discussion

The current two experiments set out to elucidate behavioural and autonomic components of defensive states by means of oculomotor, cardiovascular and electrodermal dynamics. Specifically, we pursued two major aims: first, to test whether reduced visual scanning is a stable phenomenon during the anticipation of an approaching avoidable threat, and second, to examine whether changes in gaze behaviour are suitable to discriminate between defensive and appetitive responding and whether they are adaptive for preparing subsequent threat-responses.

Supporting our first hypothesis, we replicated previously found reductions in visual scanning—denoted by decreasing average fixation distances from the centre of the screen (i.e. increased centre bias)—when participants expected an avoidable aversive stimulation (versus inevitable or no stimulation) even though threat-escape required more distributed spatial attention [19]. This pattern of reduced visual exploration upon avoidable threat was evident in horizontal and vertical components of fixations even though a broad horizontal scanning of the display—as observed in the reward context of Experiment 2—might have been advantageous to quickly detect peripherally presented action cues. With respect to our second hypothesis, we failed to find comparable changes in gaze behaviour in a rewarding context when looking at global distances of fixations from the centre of the screen. Notably though, we revealed a more nuanced picture by dissecting the distribution of fixations into horizontal and vertical components. Whereas a reduction in visual exploration was evident on the vertical axis, we observed no such effect for horizontal eye movements when participants prepared a quick movement toward an indicated side (left or right) to win a reward.

While increased skin conductance levels and enhanced pupil dilation were observed during the anticipation of inevitable and avoidable shocks as well as when participants expected an achievable reward (versus guaranteed or no reward), heart rate changes mainly differed between the active condition that required a response and the other two conditions across both experiments. Although the observed heart rate pattern consisting of an initial increase followed by a decrease and another increase starting shortly before trial outcome in Experiment 1 is consistent with the so-called cardiac defense [44] and lines up with previous findings on specific fear states in both rodents [45] and humans ([18,22,26,46], but see [29]), it was also evident on achievable reward trials in Experiment 2. Thus, contrasting the idea that transient bradycardia might be more sensitive to threat than reward processing [32], heart rate deceleration during anticipation of a motor response seems to be an important element of a more general action-preparatory mechanism independent of contextual valence, which may support processes of attentional orienting and motor preparation [31,33]. The overall pattern of skin conductance changes and pupil dilation in conjunction with heart rate deceleration indicates a co-activation of sympathetic and parasympathetic branches of the autonomic nervous system in threat and reward contexts, which suggests some resemblance of physiological responses in prey preparing to avoid harm and predators preparing for approach [32]. However, the current findings also indicate a shift toward sympathetic activity in the threat context. In this regard, it

seems important to note that we did not explicitly match the negative valence of aversive shocks with the positive valence of financial rewards. Ensuring such comparability is extremely difficult [47] but should be a matter for future research. Nevertheless, we believe that the motivational value of negative outcomes in Experiment 1 and positive outcomes in Experiment 2 was somehow comparable given that autonomic as well as behavioural responses were very similar between avoidable shock and achievable reward trials in the current study.

Importantly, the unique occurrence of globally reduced visual scanning (i.e. on both the horizontal and the vertical axes) in a threatening context, which did not merely reflect task-specific demands [37] and predicted the speed of avoidance response, highlights the adaptive and defensive nature of this oculomotor component. Whether this centralization of gaze is indeed a threat-specific component of the defensive state itself or an attentional shift as part of an anticipatory state has yet to be conclusively addressed. As oculomotor responding has previously been shown to differ between gaining rewards and risking losses [48], a direct comparison between (monetary) losses and shocks in future studies could be one way of further exploring the threat-specificity and defensive nature of this effect. Importantly, similar to reductions in body sway, the currently observed decrease of oculomotor activity can be characterized as inhibition of motion, which, due to similar temporal dynamics, allows an initial classification of these responses to reflect freezing-like behavioural states that promote fast subsequent defensive actions [18,29].

In the current study, fixation durations and numbers did just marginally differ between trial types and thereby failed to replicate previous findings [19]. Whether this was due to changes in the experimental design, low reliability of corresponding measurements or genuinely absent effects on these fixation metrics remains an important question for future research. In general, centralization of gaze may best reflect oculomotor changes during the expectation of avoidable threat and future studies should explore whether these reductions in visual exploration correlate with reduced bodily movement [7]. Moreover, it remains unclear whether and how reduced visual scanning interacts with gaze preferences evoked by more heterogeneous or dynamic visual material such as social scenes, video clips or three-dimensional virtual environments (e.g. [40,49,50]). Finally, although we are not aware of gender differences in autonomic or oculomotor dynamics during the anticipation of avoidable threat in healthy individuals, the gender imbalance in our sample might limit the generalizability of our findings to men.

In conclusion, the current study offers some important insights and implications for future research on defensive states in humans. First, we showed that a global centralization of gaze is a stable and specific phenomenon during the anticipation of avertable threat. By contrast, bradycardia alone seems to be less threat-specific but instead reflects a more general action-preparatory mechanism when a motor response is required (regarding inevitable threat, see [22]). These findings confirm that defensive responding, which integrates behavioural and autonomic functions (among other components, such as endocrine responses) bears complex temporal dynamics that need to be considered when using them as indicators of fear. Instead of relying on single output measures, integrated analyses of multiple readouts appear more appropriate to define such defense states, which seems relevant to both animal and human research. Future studies should hence consider the multifaceted nature of heart rate deceleration when adducing it as a proxy for fear-related behaviour (e.g. in neuroimaging environments [30]), particularly when creating ambivalent experimental conditions involving conflicts between threat and reward possibilities. Eye-tracking, on the other hand, seems to be more suitable to discriminate between defensive and appetitive states (see also [51]). As a whole, the current study contributes to a more comprehensive and nuanced picture of defensive states by elucidating the integrated nature of their autonomic and oculomotor components.

Ethics. The experiments were conducted according to the declaration of Helsinki and have been approved by the ethics committee of the University of Würzburg (GZEK 2019-16).

Data accessibility. The datasets generated and analysed during the current study are available on the Open Science Framework. Instructions can be found in the Readme document accessible at https://osf.io/whpt5/?view_only=34f40d237f234c24b91f3a06e2c81a24.

The data are provided in electronic supplementary material [52].

Authors' contributions. A.-S.M.: conceptualization, data curation, formal analysis, investigation, methodology, project administration, visualization, writing—original draft; P.T.: supervision, validation, visualization, writing—review and editing; P.P.: supervision, validation, writing—review and editing; M.G.: conceptualization, data curation, formal analysis, methodology, resources, software, supervision, validation, visualization, writing—review and editing.

All authors gave final approval for publication and agreed to be held accountable for the work performed therein.

Conflict of interest declaration. We declare we have no competing interests.

Funding. We received no funding for this study.

Acknowledgements. We thank Susanne Nützel and Marie Gramann for their help with data collection and Jérémy Signoret-Genest for his support in generating the figures.

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
