## [Peer Review File · Proceedings of the Royal Society B: Biological Sciences]

Review History

RSPB-2021-1656.R0 (Original submission)

Review form: Reviewer 1

Recommendation

Major revision is needed (please make suggestions in comments)

Scientific importance: Is the manuscript an original and important contribution to its field?

Acceptable

General interest: Is the paper of sufficient general interest?

Acceptable

Quality of the paper: Is the overall quality of the paper suitable?

Acceptable

Is the length of the paper justified?

Yes

Should the paper be seen by a specialist statistical reviewer?

No

Do you have any concerns about statistical analyses in this paper? If so, please specify them explicitly in your report.

Yes

It is a condition of publication that authors make their supporting data, code and materials available - either as supplementary material or hosted in an external repository. Please rate, if applicable, the supporting data on the following criteria.

Is it accessible?

N/A

Is it clear?

N/A

Is it adequate?

N/A

Do you have any ethical concerns with this paper?

No

Comments to the Author

This manuscript reports an interesting study with comprehensive assessment of responding to certain and uncertain threats and rewards. The paper has many strengths, notably the multiple measures of defensive responding (eye tracking, pupillometry, heart rate, and skin conductance measures), the authors' skill in using these methods, and their excellent data visualization. Also, the phenomenon of gaze centrality and its use as a measure of freezing are intriguing, and the authors replicate some of their group's previous findings regarding oculomotor threat responding, and extend this work by including reward in addition to threat. However, the paper also has some limitations. My largest concern is that the authors do not adequately frame the study in relation to senior author's prior work (Roesler and Gamer, 2019). This study seems to be a replication and extension of this prior study, but the introduction does not provide an adequate summary of the prior study or how specifically the present study builds upon it. Also, given that the study is framed around gaze centrality as a freezing behavior, it would be helpful if the authors discussed the function and perhaps components of scanning behavior and visual exploration, as well as their relation to defensive responding, in more depth in the introduction and discussion. This is particularly important, because it is not entirely clear that centralized gaze in the present study reflects a generalized form of defensive responding, rather than the demands of the present task. Finally, it is unclear why the threat and reward conditions are not included in the same experiment, but instead are completed as two separate experiments.

Comments:

The authors mention a previous study in this line of work (Roesler and Gamer, 2019), but they do not explain how the present study builds incrementally on this previous study, which is surprising, because they are fairly closely replicating this study in experiment 1, and then extending it by examining the effects in relation to reward in experiment 2. The introduction should have a paragraph explaining this previous study – what they did, what they found – and then explain how the present study builds incrementally upon this work. Also, in the discussion, it would be helpful if they spent more time considering the divergence between the studies. Are there some potential reasons why the other oculomotor findings (regarding number of fixations, fixation duration) from Roesler and Gamer (2019) not replicate?

Also, why was the task of Roesler and Gamer (2019) changed, in regards to the behavior that averted shock or obtained reward? In the prior task, text appeared on the screen, and participants had to press spacebar to avoid the shock; in the present task, doors or treasure chests are presented on the edges of the screen (left and right), and one of them is open – participants then move a joystick rapidly towards the open one. On my reading, this change in the task makes the gaze centrality seem more like a product of the task itself, rather than defensive responding (e.g., freezing) more generally, because fixating centrally is probably ideal for responding quickly to the doors or treasure chests presented in the periphery. Avoiding the shock requires identifying which side of the screen the open door is on, and then moving the joystick towards it. If I were a participant in this study, I would want to avoid the shock, and so on trials on which I had an opportunity to avert a shock, I would be focused on preparing for the quick response needed to avert the shock. I would likely prepare by looking at the center of the screen, which would evenly prepare me for responding to an open door on the left or right. Is this an evolved freezing response, or is it just preparing to avert a shock in the specific task situation that the authors have created? Indeed, the finding that centralized gaze predicts the speed of subsequent responding is consistent with this interpretation. I find the prior task with the space bar response better for demonstrating gaze centrality as a more generalized freezing response (i.e., not a task specific response), because the shock-averting response does not require shifting visuospatial attention, and thus there would be less need to prepare one's gaze (by fixating centrally). Still, I am not totally convinced that increasing centralized gaze prior to an upcoming visual stimulus that cues the possibility for averting shock is a feature of defensive responding that would generalize to other contexts (rather than a behavior specific to the demands of this task). This is one reason why it would be helpful to spend more time discussing visual exploration in the introduction.

Also, another possible interpretation of the results is that participants are more motivated to avoid an uncertain shock than they are to obtain an uncertain reward (hence more gaze preparation for avoidance behavior), and this likely does not reflect differences in the nature of defensive and appetitive responding, but rather absolute valence of the incentives (the shock is probably more unpleasant than the reward is pleasant).

The paragraph that begins on line 69 is very long. It could be broken into two or three paragraphs and organized a little better. It might be helpful to begin by explaining how freezing is measured in humans, particularly given that the present paper offers a new potential measure of freezing behavior (reduced sway is briefly alluded to later in the paragraph).

It would help to describe the content of the images used in the study, for those not familiar with the database. Were they affectively neutral? Did they depict landscapes and objects?

This sentence could be explained further: "The pictures were presented in a random order with half of them horizontally flipped to prevent systematic horizontal biases in eye movements provoked by basic pictorial features." What exactly is meant by horizontally flipped? What is the horizontal bias in eye movements? A citation regarding the horizontal bias would help if an appropriate one exists. I am familiar with the tendency to first look left; is this the horizontal bias, or does it refer to a general horizontal tendency in eye movements?

I found this wording confusing: "...participants were instructed whether to expect a certain (red; Exp. 1: shock condition, Exp. 2: no reward condition), no (green; Exp. 1: no shock condition, Exp. 2: reward condition) or an avoidable/achievable (yellow; Exp. 1: potential shock condition; Exp. 2: potential reward condition) aversive electrocutaneous stimulation (Experiment 1) or reward (Experiment 2) after disappearance of the naturalistic picture." Also, perhaps I am misunderstanding, but it seems like there was a fixed relation between the color of the cue and the condition. This isn't too serious of a confound, but why not randomize the color-cue pairing between participants?

Although all the durations are listed in the figure, only some are listed in the text (duration of image, which is important, is not listed in text).

I don't understand why the complicated offline drift correction procedure was used. It would seem much more straightforward (and in line with prior research) to remove trials in which participants were not fixating within some distance (e.g., 1 degree of visual angle) from the fixation cross. The authors used the EyeLink's event detection algorithms. Why not use the fixation events and their coordinates when deciding if a participant did not fixate centrally prior to a trial? Also, the offline drift correction struck me as unusual, too. It's my experience that error is not consistent across the screen. Just because the point of gaze estimation at the center for the screen is off in a certain direction by a certain distance does not mean the same error will occur at all points on the screen. I could be wrong about this, but in my experience validating the calibrations on EyeLink systems, the error is not consistent in this manner. I don't doubt that these procedures were appropriate, but with the information provided, I am left confused, and other readers might feel the same way.

This result was difficult to interpret: "A generalized linear mixed model (GLMM) with participant IDs as random intercept (see Supplementary Material for further details) demonstrated that global center bias during the second half of picture viewing predicted faster response times in potential shock trials on a trial-wise basis ($\beta = .03$, $SE = .01$, $t(875.13) = 2.32$, $p = .021$)." It might help to add a data analysis plan prior to results to explain the analyses more.

Did the authors examine correlations between the oculomotor responding of interest and the psychophysiological measures of defensive responding? I understand that these measures could have low reliability, making it difficult to observe their true relationship, but many readers would be interested in the coherence between these responses.

I did not see a description of the shock in the methods section. It would be helpful to provide more information on this feature of the methodology so that the work can be replicated by others. Perhaps the authors referred readers to the group's prior study and I missed it?

In the figures, a line denotes regions of significant difference. The authors report using some false discovery correction, but I could not find more details in the manuscript.

It is also unclear why the power analysis is for the individual experiments, and not for the critical comparison of the two conditions, which the authors ultimately conduct in their analysis. Why not randomly assign participants to the two conditions in one experiment? Also, out of my own curiosity, would it be impossible to do the two conditions (threat and reward) within subjects?

In the prior study, the authors looked at individual differences in anxiety-related traits. Did the authors repeat these analyses in the present study?

It would be helpful to report the internal consistency of the variables. I know this is difficult because the authors looked at trends across the trial. But for the eye tracking variables, I wonder, were gaze centrality, total fixations, and fixation durations consistent across trials when looking at metrics for the entire trial (rather than time points within the trial)? Reporting reliability will help readers determine if these metrics could be used in individual difference research, which would seem important given the clinical framing in the introduction. See Parsons et al. (2019):

Parsons, S., Kruijt, A. W., & Fox, E. (2019). Psychological science needs a standard practice of reporting the reliability of cognitive-behavioral measurements. *Advances in Methods and Practices in Psychological Science*, 2(4), 378-395.

Review form: Reviewer 2

Recommendation

Major revision is needed (please make suggestions in comments)

Scientific importance: Is the manuscript an original and important contribution to its field?

Excellent

General interest: Is the paper of sufficient general interest?

Excellent

Quality of the paper: Is the overall quality of the paper suitable?

Good

Is the length of the paper justified?

Yes

Should the paper be seen by a specialist statistical reviewer?

No

Do you have any concerns about statistical analyses in this paper? If so, please specify them explicitly in your report.

No

It is a condition of publication that authors make their supporting data, code and materials available - either as supplementary material or hosted in an external repository. Please rate, if applicable, the supporting data on the following criteria.

Is it accessible?

Yes

Is it clear?

Yes

Is it adequate?

Yes

Do you have any ethical concerns with this paper?

No

Comments to the Author

In this interesting study, Merscher and colleagues tested response patterns of heart rate, skin conductance and ocular measures (pupil dilation, eye movements) as indicators of defensive and/or appetitive responding. In their first experiment, autonomous and ocular responses were investigated, evoked while the experimental subjects anticipated inevitable threat of shock, the absence of threat or a threat that could be averted by active avoidance behaviour. Accordingly, in the second experiment, the shock was replaced with a monetary reward and responses were examined during the anticipation of guaranteed reward, the absence of reward or a reward that could be obtained by active approach behaviour. In either experiment there were 50 human participants.

Cardiac deceleration, skin conductance and pupil dilation increased during the anticipation of threat, which was most pronounced for heart rate and pupil dilation when active behaviour was required to avoid the shock. In the reward context, however, cardiac deceleration, skin conductance and pupil dilation seemed to be only increased during the anticipation of reward,

when it required active behaviour to be obtained. The investigation of eye movements further revealed particularly reduced visual exploration during the anticipation of evitable threat, while in anticipation of obtainable rewards, exploration was mostly unchanged. The authors, therefore, conclude that the investigated autonomous responses reflect a state of action preparation for both avoiding threat and approaching reward, while freezing of visual exploration may be an exclusive index of defensive response preparation. Correspondingly, cardiac deceleration predicted faster avoidance/approach behaviour, while freezing of visual exploration exclusively predicted faster threat avoidance.

I think the study has been elegantly conducted, is methodologically sound and, in general, well analysed. The results are very important to a broad audience of basic and translational neuroscientists, as they expand our knowledge of the functional aspects and expressions of defensive and appetitive responding. I therefore believe, that this manuscript will arouse much interest in the scientific community. However, I have a few concerns that need to be addressed before the manuscript may be considered ready for publication.

Abstract

a) The authors mainly focus on the response differences during anticipation of threat, that requires action to be avoided (active threat), vs. the anticipation of reward, that requires action to be obtained (active reward). However, the other experimental contexts are of interest as well, helping to link the current data to previous literature in the field (e.g., Löw et al., 2008 and 2015). The authors, therefore, should also include their findings on defensive/appetitive responding during anticipation of inevitable threat/guaranteed reward in the abstract and, based on these findings, outline how responses in the active conditions differed.

b) The conditions during which action was required for threat avoidance/reward are termed “potential threat” or “potential reward”, respectively. As far as I’m aware of, the term “potential” has primarily been used in past research to highlight that the occurrence of threat/reward is unpredictable, but inevitable (see Davis et al., 2010 or Schmitz et al., 2011). The authors should therefore rephrase the potential condition either to active threat/active reward (as done in Löw et al., 2015) or evitable threat/obtainable reward to avoid misunderstandings. This would also apply to the entire paper.

c) It should be stated, that there were two experiments

d) Only a minor suggestion: The authors may want to consider to focus less on defensive responding in the abstract, but frame the findings in a predator-prey/defense-approach context, as previously done (see Löw et al., 2008 or Lang et al., 1997) – i.e., that physiological responses in preys preparing for avoidance resemble those in predators preparing for approach, providing an excellent link to the study. This may also help to broaden the scope of the manuscript, highlight the novelty of the findings and enhance the impact in the scientific community. This would also apply for the introduction and the discussion.

Introduction

a) My major concern is, that the concept of freezing is not elaborated in enough detail, which made it hard to follow the author’s thoughts. Introducing attentive immobility/attentive freezing in the context of the dynamic organization of defensive responding (see Hamm, 2020) may provide a more profound basis for the study’s design. It may also help to increase comprehensibility of the introduction. Here, attentive immobility renders a state of vigilant readiness during confrontations with inevitable but rather distal threat, where the organism monitors the source of danger and prepares for defensive action, if the threat becomes more imminent or an avoidance option eventually opens up, which resembles the situation of the participants in the current study (see also Lang, 2000). Cardiac deceleration (so called fear bradycardia) as well as high responsivity to external events are hallmarks of this defensive state, while during more imminent threat stages, when defensive action is required, heart rate supportively accelerates, which resembles the 1-s action period in the current study. Furthermore, attentive immobility may, thus, be separated from tonic immobility, evoked during imminent threat stages and associated with cardiac acceleration and unresponsiveness (see Volchan et al.,

2017). The authors may also be interested in further literature on the relationship between heart rate and attentive immobility (Szeska et al., 2021).

b) Based on the previous comment, I disagree with the general assumption of the authors, that reduced body sway in passive contexts – which are reliably associated with cardiac decelerations – reflect tonic immobility (lines 73 and 74)

c) The authors may also want to consider implementing previous research by Löw and colleagues (2008), where changes in skin conductance and heart rate were also measured at varying stages of threat/reward imminence, including stages of action preparation for avoidance/approach.

Methods

a) The authors should specify, what was problematic about the eye tracking data or made the heart rate invalid, that justified an exclusion of the respective participants from the analyses. This is important, especially given the current controversy on excluding participants on the basis of response measures (see Lonsdorf et al., 2019)

b) The 8-s duration of the presentation of the naturalistic pictures should also be mentioned in the text, not only in Figure 1

c) I missed sections, where the electrical shock is described, especially whether the intensity was fixed or individually adjusted. If the latter is the case (which I assume, since has been done in the first investigation from 2019), the authors should provide some details on the shock workup at least in the supplemental material.

d) A short sentence at the end of the methods should provide some general details on the statistical analyses and refer to the supplements for further reading.

Results

a) Similar to the abstract, my major concern with the results is that the authors primarily focused on distinguishing the active threat/reward condition from the other two. Instead, in my view, it would be better to first characterize how the certain threat/certain reward condition differed from the no threat/no reward condition and then report, how the active conditions evoked different response patterns. In other words: The authors should disentangle the interactions they found in more detail. This may also answer the following questions I had while reading the results section:

b) Figure 4a seems to indicate, that cardiac deceleration gradually increases from no-threat to certain-threat to avoidable-threat, primarily during seconds 6-8, which circumscribes the time of the D2-component of cardiac responding, suggested to indicate threat anticipation (see Szeska et al., 2021). As, again, the authors only report that cardiac deceleration differed significantly between the active-threat condition and the other two, I was wondering if this impression proves right?

c) In contrast, Figure 4d seems to indicate, that while cardiac deceleration was particularly pronounced in the active reward condition, the other two reward conditions did not differ. Is that right?

d) If b) and c) are true and especially if cardiac deceleration is additionally stronger in certain shock compared to certain reward trials, as might be concluded from the significant three-way-interaction (lines 278-283) and a comparison of Figures 4a and d, this would indicate that cardiac deceleration is indeed more specific/more sensitive to processing of threat relative to processing of reward. In that case, the discussion should be rephrased.

e) The authors should at least mention the profound cardiac acceleration at the end of picture viewing in both active conditions, which might mark the transition from attentive immobility to active avoidance/approach. Given that, the cardiac acceleration at the end of the certain threat condition should also be mentioned and discussed later.

Discussion

a) As for the introduction, the authors should discuss the results also in the context of Löw et al. (2008), who also investigated autonomous measures during threat and reward anticipation.

b) Line 339: The authors should be cautious at suggesting, that cardiac deceleration is independent of contextual valence. Löw et al. (2008) found stronger cardiac deceleration during

threat anticipation relative to reward anticipation.

- c) Do the authors have any explanation for the failed replication (lines 356-358)?
- d) Given the results from Szeska et al. (2021), the authors may want to tone down the conclusion drawn in lines 393-396

Decision letter (RSPB-2021-1656.R0)

02-Sep-2021

Dear Miss Merscher:

I have now received two reviews of your manuscript RSPB-2021-1656 entitled "Centralized gaze as a threat-specific component of defensive states in humans" has, in its current form, been rejected for publication in Proceedings B. Based on their reviews and the assessment of the Associate editor, as well as my own read of your manuscript, I am rejecting it in its current form. However, all four of us appreciate your study and see potential, so I am inviting you to submit a revised version if you are able to address each of the reviewers' concerns. The reviewers do a particularly thorough job of outlining their comments, which are appended below, so I will not go over them again in detail, but I do highlight the necessity of including all of the details and ensuring that it is clear how your manuscript is related to - and distinct from - earlier work. In addition, I agree with the AE's comments about the gender imbalance in your sample. I recognize the irony that it is typically a male dominated sample in such cases, but if you are unable to collect data from additional male participants please be sure to address this and consider the potential ramifications. Please also report other relevant characteristics of your study population (are they from university, whether they were vision corrected, tested for color blindness, etc). Finally, reviewer 2 makes some suggestions for framing that would increase the breadth of scope of your manuscript that I encourage you to consider, particularly as Proceedings' mandate is to publish manuscripts of broad biological relevance. Once we have received your revised manuscript we will send it back out to the original reviewers, should they be available, and please note that this is not a provisional acceptance.

Sincerely,
Dr Sarah Brosnan
Editor, Proceedings B
mailto: proceedingsb@royalsociety.org

Associate Editor
Comments to Author:

This is an interesting and multifaceted study that I believe offers important and novel insights. Two reviewers have provided feedback on this article and both agree that it has merit, but both also provide detailed and thoughtful suggestions as to how the reporting of this study can be enhanced, both with respect to the framing and the clarity of the methodological approach. A careful response to their feedback would greatly enhance this article I believe.

In addition to the reviewers' feedback, I also have some concerns, some of which echo the comments made by the reviewers.

First, and as noted by the authors in this article, this study represents an extension of a previously published study by Roesler and Gamer (2019). Both in the Introduction and Methods, the authors must explicitly outline how this new study differs from the previous one and what novel insights this new approach offers i.e. what novel insights might this new study offer to the field?

Second, while a reasonable number of participants were included in both studies (N=50), 80% of the participants were female. What is the reason for this skew? Ideally additional male participants would be tested. At the very least, the authors must acknowledge and discuss this potential limitation in the generalizability of their data set. Related to this, it would be helpful if the authors provided greater detail about the people they tested i.e. their familiarity with such testing protocols (e.g., were they psychology students) and if any were colorblind which may have influenced their perception of the stimuli.

Third, the description of the methods lacked key detail and could not be replicated as described. Specifically related to the testing set up. For example, where the participants tested alone in a room? were other people present? what were the lighting conditions? was it the same room for all participants? Such factors will likely influence the response of the participants and also the accuracy of the eye tracking measures. Such information can be provided as supplementary materials, but must be included.

Lastly, and as noted by the reviewers, the analytical approach should be presented in the body of the article. How you analyze your data is key to understand for the interpretation of the results.

Reviewer(s)' Comments to Author:

Referee: 1

Comments to the Author(s)

This manuscript reports an interesting study with comprehensive assessment of responding to certain and uncertain threats and rewards. The paper has many strengths, notably the multiple measures of defensive responding (eye tracking, pupillometry, heart rate, and skin conductance measures), the authors' skill in using these methods, and their excellent data visualization. Also, the phenomenon of gaze centrality and its use as a measure of freezing are intriguing, and the authors replicate some of their group's previous findings regarding oculomotor threat responding, and extend this work by including reward in addition to threat. However, the paper also has some limitations. My largest concern is that the authors do not adequately frame the study in relation to senior author's prior work (Roesler and Gamer, 2019). This study seems to be a replication and extension of this prior study, but the introduction does not provide an adequate summary of the prior study or how specifically the present study builds upon it. Also, given that

the study is framed around gaze centrality as a freezing behavior, it would be helpful if the authors discussed the function and perhaps components of scanning behavior and visual exploration, as well as their relation to defensive responding, in more depth in the introduction and discussion. This is particularly important, because it is not entirely clear that centralized gaze in the present study reflects a generalized form of defensive responding, rather than the demands of the present task. Finally, it is unclear why the threat and reward conditions are not included in the same experiment, but instead are completed as two separate experiments.

Comments:

The authors mention a previous study in this line of work (Roesler and Gamer, 2019), but they do not explain how the present study builds incrementally on this previous study, which is surprising, because they are fairly closely replicating this study in experiment 1, and then extending it by examining the effects in relation to reward in experiment 2. The introduction should have a paragraph explaining this previous study – what they did, what they found – and then explain how the present study builds incrementally upon this work. Also, in the discussion, it would be helpful if they spent more time considering the divergence between the studies. Are there some potential reasons why the other oculomotor findings (regarding number of fixations, fixation duration) from Roesler and Gamer (2019) not replicate?

Also, why was the task of Roesler and Gamer (2019) changed, in regards to the behavior that averted shock or obtained reward? In the prior task, text appeared on the screen, and participants had to press spacebar to avoid the shock; in the present task, doors or treasure chests are presented on the edges of the screen (left and right), and one of them is open – participants then move a joystick rapidly towards the open one. On my reading, this change in the task makes the gaze centrality seem more like a product of the task itself, rather than defensive responding (e.g., freezing) more generally, because fixating centrally is probably ideal for responding quickly to the doors or treasure chests presented in the periphery. Avoiding the shock requires identifying which side of the screen the open door is on, and then moving the joystick towards it. If I were a participant in this study, I would want to avoid the shock, and so on trials on which I had an opportunity to avert a shock, I would be focused on preparing for the quick response needed to avert the shock. I would likely prepare by looking at the center of the screen, which would evenly prepare me for responding to an open door on the left or right. Is this an evolved freezing response, or is it just preparing to avert a shock in the specific task situation that the authors have created? Indeed, the finding that centralized gaze predicts the speed of subsequent responding is consistent with this interpretation. I find the prior task with the space bar response better for demonstrating gaze centrality as a more generalized freezing response (i.e., not a task specific response), because the shock-averting response does not require shifting visuospatial attention, and thus there would be less need to prepare one's gaze (by fixating centrally). Still, I am not totally convinced that increasing centralized gaze prior to an upcoming visual stimulus that cues the possibility for averting shock is a feature of defensive responding that would generalize to other contexts (rather than a behavior specific to the demands of this task). This is one reason why it would be helpful to spend more time discussing visual exploration in the introduction.

Also, another possible interpretation of the results is that participants are more motivated to avoid an uncertain shock than they are to obtain an uncertain reward (hence more gaze preparation for avoidance behavior), and this likely does not reflect differences in the nature of defensive and appetitive responding, but rather absolute valence of the incentives (the shock is probably more unpleasant than the reward is pleasant).

The paragraph that begins on line 69 is very long. It could be broken into two or three paragraphs and organized a little better. It might be helpful to begin by explaining how freezing is measured in humans, particularly given that the present paper offers a new potential measure of freezing behavior (reduced sway is briefly alluded to later in the paragraph).

It would help to describe the content of the images used in the study, for those not familiar with the database. Were they affectively neutral? Did they depict landscapes and objects?

This sentence could be explained further: “The pictures were presented in a random order with half of them horizontally flipped to prevent systematic horizontal biases in eye movements provoked by basic pictorial features.” What exactly is meant by horizontally flipped? What is the horizontal bias in eye movements? A citation regarding the horizontal bias would help if an appropriate one exists. I am familiar with the tendency to first look left; is this the horizontal bias, or does it refer to a general horizontal tendency in eye movements?

I found this wording confusing: “...participants were instructed whether to expect a certain (red; Exp. 1: shock condition, Exp. 2: no reward condition), no (green; Exp. 1: no shock condition, Exp. 2: reward condition) or an avoidable/achievable (yellow; Exp. 1: potential shock condition; Exp. 2: potential reward condition) aversive electrocutaneous stimulation (Experiment 1) or reward (Experiment 2) after disappearance of the naturalistic picture.” Also, perhaps I am misunderstanding, but it seems like there was a fixed relation between the color of the cue and the condition. This isn’t too serious of a confound, but why not randomize the color-cue pairing between participants?

Although all the durations are listed in the figure, only some are listed in the text (duration of image, which is important, is not listed in text).

I don’t understand why the complicated offline drift correction procedure was used. It would seem much more straightforward (and in line with prior research) to remove trials in which participants were not fixating within some distance (e.g., 1 degree of visual angle) from the fixation cross. The authors used the EyeLink’s event detection algorithms. Why not use the fixation events and their coordinates when deciding if a participant did not fixate centrally prior to a trial? Also, the offline drift correction struck me as unusual, too. It’s my experience that error is not consistent across the screen. Just because the point of gaze estimation at the center for the screen is off in a certain direction by a certain distance does not mean the same error will occur at all points on the screen. I could be wrong about this, but in my experience validating the calibrations on EyeLink systems, the error is not consistent in this manner. I don’t doubt that these procedures were appropriate, but with the information provided, I am left confused, and other readers might feel the same way.

This result was difficult to interpret: “A generalized linear mixed model (GLMM) with participant IDs as random intercept (see Supplementary Material for further details) demonstrated that global center bias during the second half of picture viewing predicted faster response times in potential shock trials on a trial-wise basis ($\beta = .03$, $SE = .01$, $t(875.13) = 2.32$, $p = .021$).” It might help to add a data analysis plan prior to results to explain the analyses more.

Did the authors examine correlations between the oculomotor responding of interest and the psychophysiological measures of defensive responding? I understand that these measures could have low reliability, making it difficult to observe their true relationship, but many readers would be interested in the coherence between these responses.

I did not see a description of the shock in the methods section. It would be helpful to provide more information on this feature of the methodology so that the work can be replicated by others. Perhaps the authors referred readers to the group’s prior study and I missed it?

In the figures, a line denotes regions of significant difference. The authors report using some false discovery correction, but I could not find more details in the manuscript.

It is also unclear why the power analysis is for the individual experiments, and not for the critical comparison of the two conditions, which the authors ultimately conduct in their analysis. Why not randomly assign participants to the two conditions in one experiment? Also, out of my own curiosity, would it be impossible to do the two conditions (threat and reward) within subjects?

In the prior study, the authors looked at individual differences in anxiety-related traits. Did the authors repeat these analyses in the present study?

It would be helpful to report the internal consistency of the variables. I know this is difficult because the authors looked at trends across the trial. But for the eye tracking variables, I wonder, were gaze centrality, total fixations, and fixation durations consistent across trials when looking at metrics for the entire trial (rather than time points within the trial)? Reporting reliability will help readers determine if these metrics could be used in individual difference research, which would seem important given the clinical framing in the introduction. See Parsons et al. (2019):

Parsons, S., Kruijt, A. W., & Fox, E. (2019). Psychological science needs a standard practice of reporting the reliability of cognitive-behavioral measurements. *Advances in Methods and Practices in Psychological Science*, 2(4), 378-395.

Referee: 2

Comments to the Author(s)

In this interesting study, Merscher and colleagues tested response patterns of heart rate, skin conductance and ocular measures (pupil dilation, eye movements) as indicators of defensive and/or appetitive responding. In their first experiment, autonomous and ocular responses were investigated, evoked while the experimental subjects anticipated inevitable threat of shock, the absence of threat or a threat that could be averted by active avoidance behaviour. Accordingly, in the second experiment, the shock was replaced with a monetary reward and responses were examined during the anticipation of guaranteed reward, the absence of reward or a reward that could be obtained by active approach behaviour. In either experiment there were 50 human participants.

Cardiac deceleration, skin conductance and pupil dilation increased during the anticipation of threat, which was most pronounced for heart rate and pupil dilation when active behaviour was required to avoid the shock. In the reward context, however, cardiac deceleration, skin conductance and pupil dilation seemed to be only increased during the anticipation of reward, when it required active behaviour to be obtained. The investigation of eye movements further revealed particularly reduced visual exploration during the anticipation of evitable threat, while in anticipation of obtainable rewards, exploration was mostly unchanged. The authors, therefore, conclude that the investigated autonomous responses reflect a state of action preparation for both avoiding threat and approaching reward, while freezing of visual exploration may be an exclusive index of defensive response preparation. Correspondingly, cardiac deceleration predicted faster avoidance/approach behaviour, while freezing of visual exploration exclusively predicted faster threat avoidance.

I think the study has been elegantly conducted, is methodologically sound and, in general, well analysed. The results are very important to a broad audience of basic and translational neuroscientists, as they expand our knowledge of the functional aspects and expressions of defensive and appetitive responding. I therefore believe, that this manuscript will arouse much interest in the scientific community. However, I have a few concerns that need to be addressed before the manuscript may be considered ready for publication.

Abstract

a) The authors mainly focus on the response differences during anticipation of threat, that requires action to be avoided (active threat), vs. the anticipation of reward, that requires action to be obtained (active reward). However, the other experimental contexts are of interest as well, helping to link the current data to previous literature in the field (e.g., Löw et al., 2008 and 2015). The authors, therefore, should also include their findings on defensive/appetitive responding during anticipation of inevitable threat/guaranteed reward in the abstract and, based on these findings, outline how responses in the active conditions differed.

b) The conditions during which action was required for threat avoidance/reward are termed “potential threat” or “potential reward”, respectively. As far as I’m aware of, the term “potential” has primarily been used in past research to highlight that the occurrence of threat/reward is unpredictable, but inevitable (see Davis et al., 2010 or Schmitz et al., 2011). The authors should therefore rephrase the potential condition either to active threat/active reward (as done in Löw et al., 2015) or evitable threat/obtainable reward to avoid misunderstandings. This would also apply to the entire paper.

c) It should be stated, that there were two experiments

d) Only a minor suggestion: The authors may want to consider to focus less on defensive responding in the abstract, but frame the findings in a predator-prey/defense-approach context, as previously done (see Löw et al., 2008 or Lang et al., 1997) – i.e., that physiological responses in preys preparing for avoidance resemble those in predators preparing for approach, providing an excellent link to the study. This may also help to broaden the scope of the manuscript, highlight the novelty of the findings and enhance the impact in the scientific community. This would also apply for the introduction and the discussion.

Introduction

a) My major concern is, that the concept of freezing is not elaborated in enough detail, which made it hard to follow the author’s thoughts. Introducing attentive immobility/attentive freezing in the context of the dynamic organization of defensive responding (see Hamm, 2020) may provide a more profound basis for the study’s design. It may also help to increase comprehensibility of the introduction. Here, attentive immobility renders a state of vigilant readiness during confrontations with inevitable but rather distal threat, where the organism monitors the source of danger and prepares for defensive action, if the threat becomes more imminent or an avoidance option eventually opens up, which resembles the situation of the participants in the current study (see also Lang, 2000). Cardiac deceleration (so called fear bradycardia) as well as high responsivity to external events are hallmarks of this defensive state, while during more imminent threat stages, when defensive action is required, heart rate supportively accelerates, which resembles the 1-s action period in the current study. Furthermore, attentive immobility may, thus, be separated from tonic immobility, evoked during imminent threat stages and associated with cardiac acceleration and unresponsiveness (see Volchan et al., 2017). The authors may also be interested in further literature on the relationship between heart rate and attentive immobility (Szeska et al., 2021).

b) Based on the previous comment, I disagree with the general assumption of the authors, that reduced body sway in passive contexts – which are reliably associated with cardiac decelerations – reflect tonic immobility (lines 73 and 74)

c) The authors may also want to consider implementing previous research by Löw and colleagues (2008), where changes in skin conductance and heart rate were also measured at varying stages of threat/reward imminence, including stages of action preparation for avoidance/approach.

Methods

a) The authors should specify, what was problematic about the eye tracking data or made the heart rate invalid, that justified an exclusion of the respective participants from the analyses. This is important, especially given the current controversy on excluding participants on the basis of response measures (see Lonsdorf et al., 2019)

b) The 8-s duration of the presentation of the naturalistic pictures should also be mentioned in the text, not only in Figure 1

c) I missed sections, where the electrical shock is described, especially whether the intensity was fixed or individually adjusted. If the latter is the case (which I assume, since has been done in the first investigation from 2019), the authors should provide some details on the shock workup at least in the supplemental material.

d) A short sentence at the end of the methods should provide some general details on the statistical analyses and refer to the supplements for further reading.

Results

a) Similar to the abstract, my major concern with the results is that the authors primarily focused on distinguishing the active threat/reward condition from the other two. Instead, in my view, it would be better to first characterize how the certain threat/certain reward condition differed from the no threat/no reward condition and then report, how the active conditions evoked different response patterns. In other words: The authors should disentangle the interactions they found in more detail. This may also answer the following questions I had while reading the results section:

b) Figure 4a seems to indicate, that cardiac deceleration gradually increases from no-threat to certain-threat to avoidable-threat, primarily during seconds 6-8, which circumscribes the time of the D2-component of cardiac responding, suggested to indicate threat anticipation (see Szeska et al., 2021). As, again, the authors only report that cardiac deceleration differed significantly between the active-threat condition and the other two, I was wondering if this impression proves right?

c) In contrast, Figure 4d seems to indicate, that while cardiac deceleration was particularly pronounced in the active reward condition, the other two reward conditions did not differ. Is that right?

d) If b) and c) are true and especially if cardiac deceleration is additionally stronger in certain shock compared to certain reward trials, as might be concluded from the significant three-way-interaction (lines 278-283) and a comparison of Figures 4a and d, this would indicate that cardiac deceleration is indeed more specific/more sensitive to processing of threat relative to processing of reward. In that case, the discussion should be rephrased.

e) The authors should at least mention the profound cardiac acceleration at the end of picture viewing in both active conditions, which might mark the transition from attentive immobility to active avoidance/approach. Given that, the cardiac acceleration at the end of the certain threat condition should also be mentioned and discussed later.

Discussion

a) As for the introduction, the authors should discuss the results also in the context of Löw et al. (2008), who also investigated autonomous measures during threat and reward anticipation.

b) Line 339: The authors should be cautious at suggesting, that cardiac deceleration is independent of contextual valence. Löw et al. (2008) found stronger cardiac deceleration during threat anticipation relative to reward anticipation.

c) Do the authors have any explanation for the failed replication (lines 356-358)?

d) Given the results from Szeska et al. (2021), the authors may want to tone down the conclusion drawn in lines 393-396

Author's Response to Decision Letter for (RSPB-2021-1656.R0)

See Appendix A.

RSPB-2022-0405.R0

Review form: Reviewer 1

Recommendation

Accept as is

Scientific importance: Is the manuscript an original and important contribution to its field?

Good

General interest: Is the paper of sufficient general interest?

Good

Quality of the paper: Is the overall quality of the paper suitable?

Good

Is the length of the paper justified?

Yes

Should the paper be seen by a specialist statistical reviewer?

No

Do you have any concerns about statistical analyses in this paper? If so, please specify them explicitly in your report.

No

It is a condition of publication that authors make their supporting data, code and materials available - either as supplementary material or hosted in an external repository. Please rate, if applicable, the supporting data on the following criteria.

Is it accessible?

N/A

Is it clear?

N/A

Is it adequate?

N/A

Do you have any ethical concerns with this paper?

Yes

Comments to the Author

The authors did a great job addressing my questions and concerns.

Review form: Reviewer 2

Recommendation

Accept as is

Scientific importance: Is the manuscript an original and important contribution to its field?

Excellent

General interest: Is the paper of sufficient general interest?

Excellent

Quality of the paper: Is the overall quality of the paper suitable?

Excellent

Is the length of the paper justified?

Yes

Should the paper be seen by a specialist statistical reviewer?

No

Do you have any concerns about statistical analyses in this paper? If so, please specify them explicitly in your report.

Yes

It is a condition of publication that authors make their supporting data, code and materials available - either as supplementary material or hosted in an external repository. Please rate, if applicable, the supporting data on the following criteria.

Is it accessible?

Yes

Is it clear?

Yes

Is it adequate?

Yes

Do you have any ethical concerns with this paper?

No

Comments to the Author

With the thorough changes by the authors, the manuscript has become more concise and the results can be evaluated more clearly. Perhaps even more important, however, the integration of the current work into a bigger defense/approach framework has - in my view - significantly improved the manuscript, which will now serve as a spark for future integrative psychophysiological research on motivated responding and emotional reactivity. All my comments have been appropriately addressed.

I want to thank the authors for the extensive responses, but even more for the intriguing manuscript, that will surely make an important contribution to the field.

Review form: Reviewer 3 (Edwin S. Dalmaijer)

Recommendation

Accept with minor revision (please list in comments)

Scientific importance: Is the manuscript an original and important contribution to its field?

Good

General interest: Is the paper of sufficient general interest?

Acceptable

Quality of the paper: Is the overall quality of the paper suitable?

Good

Is the length of the paper justified?

Yes

Should the paper be seen by a specialist statistical reviewer?

No

Do you have any concerns about statistical analyses in this paper? If so, please specify them explicitly in your report.

No

It is a condition of publication that authors make their supporting data, code and materials available - either as supplementary material or hosted in an external repository. Please rate, if applicable, the supporting data on the following criteria.

Is it accessible?

Yes

Is it clear?

Yes

Is it adequate?

Yes

Do you have any ethical concerns with this paper?

No

Comments to the Author

Centralized gaze as a threat-specific component of defensive states in humans

Merscher, Tovote, Pauli, & Gamer

OVERVIEW

=====

It seems that I have been asked to review a revised version of a manuscript, and I am painfully aware that sometimes this can result in double (or worse: completely contradictory!) comments. I have gone through the manuscript with this in mind, and do hope that the authors forgive me any inconveniences caused as a consequence of the situation.

In general, I thought this was an interesting manuscript. The authors present the idea that "freezing"-like responses could be preserved in humans in the oculomotor system, and do indeed seem to find evidence for this - but only when a potential reward or threat was contingent on subsequent action.

I am not sure whether I agree with the authors' chosen title and framing of "centralised gaze as a threat-specific component of defensive states". This is for two main reasons, outlined below.

Signed,
Edwin Dalmaijer

MAIN POINTS

=====

1) Centralised gaze was not threat-specific, but also appeared in reward. I appreciate that the authors show that this was to lessened extent in horizontal directions, and I buy into their interpretation that this was a strategy by participants to look towards the potential location of stimuli. It's nicely analogous to goal and sign tracking in rodent experiments.

The authors argue that if the centralised gaze before threat was strategic, one would also see such a response in the reward experiment. Instead, they see a different strategy. This glosses over the

fact that the oculomotor system responds differently to gaining something beneficial compared to risking something adverse. For example, Muhammed and colleagues (2020, <https://doi.org/10.1016/j.cortex.2018.12.001>) showed that saccadic velocity can speed up for contingent rewards, but not for contingent losses.

Obviously, being shocked is not the same as a monetary loss. However, I would argue that avoiding a potential loss is a better control for Experiment 1's avoiding an adverse stimulus. Without more controls, I'm not entirely convinced of the strong claim of centralised gaze being fully threat-specific.

One potential remedy would be to run an extra experiment with losses instead of rewards. This comes with the obvious downside of being a huge effort, especially in times of COVID. I don't think this type of time/effort investment is warranted. Instead, I would argue that the authors should stress the limitations of their control experiment, and to tone down their claims of central gaze being threat-specific.

2) It is somewhat unclear to me whether the gaze is a component of defensive states, or rather a by-effect of attentional processes. I realise that this sounds like a comment brought up by Reviewer 1, but I would like to elaborate on why I'm unsure even after reading the authors' responses to them.

Humans show strong attentional capture for threat. This is true when conditioned stimuli are used as task-irrelevant distractors (Mulckhuyse & Dalmaijer, 2016, <http://dx.doi.org/10.3758/s13415-015-0391-2>). In fact, oculomotor bias for threat-associated stimuli persists after successful extinction; even when self-reported expectancy and pupil dilation have reduced back to baseline (Armstrong et al., 2022, <https://doi.org/10.31234/osf.io/4ct8x>).

With the above in mind, it seems plausible participants were simply focussed on the threat-related movement they were about to make. Is this a "defensive state", or is it attention turning inwards in preparation for a movement? Or is the attentional shift a part of the "anticipatory defensive state"?

It might be due to a word limit, but the Discussion currently reads as though the authors are trying to sell their results as identifying a unique element of threat avoidance, while alternative or more nuanced explanations aren't fully explored.

NOTE: The cited studies are all studies that I was involved in. I'm not trying to be a typical Reviewer 2 here: the point is not to convince the authors to cite my work. These are simply the first studies that come to mind for me, as I have worked on them; there are similar and related papers in the literature.

MINOR POINTS

=====

Very minor point, but the authors copied the Figure 2 legend into Figure 4, but it seems they forgot to include the shock/money icons in the actual figure. (The dotted line for their onset is still visible.)

Decision letter (RSPB-2022-0405.R0)

14-Apr-2022

Dear Miss Merscher

I am pleased to inform you that your manuscript RSPB-2022-0405 entitled "Centralized gaze as a threat-specific component of defensive states in humans" has been accepted for publication in Proceedings B pending minor revisions. Therefore, I invite you to respond to the referee(s)' comments and revise your manuscript. Because the schedule for publication is very tight, it is a condition of publication that you submit the revised version of your manuscript within 7 days. If you do not think you will be able to meet this date please let us know.

When submitting your revision please upload a file under "Response to Referees" - in the "File Upload" section. This should document, point by point, how you have responded to the reviewers' and Editors' comments, and the adjustments you have made to the manuscript. We also require a copy of the revised manuscript showing track changes to be uploaded.

4) Data accessibility section and data citation

It is a condition of publication that data supporting your paper are made available either in the electronic supplementary material. Authors must complete the 'data accessibility' section in the submission system. This should list the database and accession number for all data from the article that has been made publicly available, for instance:

NB. From April 1 2013, peer reviewed articles based on research funded wholly or partly by RCUK must include, if applicable, a statement on how the underlying research materials – such as data, samples or models – can be accessed.

If you wish to submit your data to Dryad (<http://datadryad.org/>) and have not already done so you can submit your data via this link [http://datadryad.org/submit?journalID=RSPB&manu=\(Document not available\)](http://datadryad.org/submit?journalID=RSPB&manu=(Document%20not%20available)) which will take you to your unique entry in the Dryad repository. If you have already submitted your data to dryad you can make any necessary revisions to your dataset by following the above link.

Please include the Dryad DOI in the Data Accessibility section and reference in the paper's bibliography.

Please see our Data Sharing Policies (<https://royalsociety.org/journals/authors/author-guidelines/>).

6) A media summary: a short non-technical summary (up to 100 words) of the key findings/importance of your manuscript.

Sincerely,
Dr Sarah Brosnan
Editor, Proceedings B
mailto: proceedingsb@royalsociety.org

Associate Editor

Comments to Author:

The two reviewers who reviewed your original submission have now reviewed this resubmission. Both are in agreement that you have carefully responded to their original feedback and both praise the revisions that you have made. I concur. I find your revised article to be much more clear and compelling. In addition to these reviewers a third (new) reviewer has also provided feedback on this version of your article. They provide some feedback on the framing of your results that I believe you should be able to address relatively easily (while they suggest that additional controls might be beneficial, I agree with them that they are not necessary.)

Reviewer(s)' Comments to Author:

Referee: 2

Comments to the Author(s).

With the thorough changes by the authors, the manuscript has become more concise and the results can be evaluated more clearly. Perhaps even more important, however, the integration of the current work into a bigger defense/approach framework has - in my view - significantly improved the manuscript, which will now serve as a spark for future integrative psychophysiological research on motivated responding and emotional reactivity. All my comments have been appropriately addressed.

I want to thank the authors for the extensive responses, but even more for the intriguing manuscript, that will surely make an important contribution to the field.

Referee: 1

Comments to the Author(s).

The authors did a great job addressing my questions and concerns.

Referee: 3

Comments to the Author(s).

Centralized gaze as a threat-specific component of defensive states in humans

Merscher, Tovote, Pauli, & Gamer

OVERVIEW

=====

It seems that I have been asked to review a revised version of a manuscript, and I am painfully aware that sometimes this can result in double (or worse: completely contradictory!) comments. I have gone through the manuscript with this in mind, and do hope that the authors forgive me any inconveniences caused as a consequence of the situation.

In general, I thought this was an interesting manuscript. The authors present the idea that "freezing"-like responses could be preserved in humans in the oculomotor system, and do indeed seem to find evidence for this - but only when a potential reward or threat was contingent on subsequent action.

I am not sure whether I agree with the authors' chosen title and framing of "centralised gaze as a threat-specific component of defensive states". This is for two main reasons, outlined below.

Signed,
Edwin Dalmaijer

MAIN POINTS

=====

1) Centralised gaze was not threat-specific, but also appeared in reward. I appreciate that the authors show that this was to lessened extent in horizontal directions, and I buy into their interpretation that this was a strategy by participants to look towards the potential location of stimuli. It's nicely analogous to goal and sign tracking in rodent experiments.

The authors argue that if the centralised gaze before threat was strategic, one would also see such a response in the reward experiment. Instead, they see a different strategy. This glosses over the fact that the oculomotor system responds differently to gaining something beneficial compared to risking something adverse. For example, Muhammed and colleagues (2020, <https://doi.org/10.1016/j.cortex.2018.12.001>) showed that saccadic velocity can speed up for contingent rewards, but not for contingent losses.

Obviously, being shocked is not the same as a monetary loss. However, I would argue that avoiding a potential loss is a better control for Experiment 1's avoiding an adverse stimulus. Without more controls, I'm not entirely convinced of the strong claim of centralised gaze being fully threat-specific.

One potential remedy would be to run an extra experiment with losses instead of rewards. This comes with the obvious downside of being a huge effort, especially in times of COVID. I don't think this type of time/effort investment is warranted. Instead, I would argue that the authors should stress the limitations of their control experiment, and to tone down their claims of central gaze being threat-specific.

2) It is somewhat unclear to me whether the gaze is a component of defensive states, or rather a by-effect of attentional processes. I realise that this sounds like a comment brought up by Reviewer 1, but I would like to elaborate on why I'm unsure even after reading the authors' responses to them.

Humans show strong attentional capture for threat. This is true when conditioned stimuli are used as task-irrelevant distractors (Mulckhuyse & Dalmaijer, 2016, <http://dx.doi.org/10.3758/s13415-015-0391-2>). In fact, oculomotor bias for threat-associated stimuli persists after successful extinction; even when self-reported expectancy and pupil dilation have reduced back to baseline (Armstrong et al., 2022, <https://doi.org/10.31234/osf.io/4ct8x>).

With the above in mind, it seems plausible participants were simply focussed on the threat-related movement they were about to make. Is this a "defensive state", or is it attention turning inwards in preparation for a movement? Or is the attentional shift a part of the "anticipatory defensive state"?

It might be due to a word limit, but the Discussion currently reads as though the authors are trying to sell their results as identifying a unique element of threat avoidance, while alternative or more nuanced explanations aren't fully explored.

NOTE: The cited studies are all studies that I was involved in. I'm not trying to be a typical Reviewer 2 here: the point is not to convince the authors to cite my work. These are simply the first studies that come to mind for me, as I have worked on them; there are similar and related papers in the literature.

MINOR POINTS

=====

Very minor point, but the authors copied the Figure 2 legend into Figure 4, but it seems they forgot to include the shock/money icons in the actual figure. (The dotted line for their onset is still visible.)

Author's Response to Decision Letter for (RSPB-2022-0405.R0)

See Appendix B.

Decision letter (RSPB-2022-0405.R1)

22-Apr-2022

Dear Miss Merscher

I am pleased to inform you that your manuscript entitled "Centralized gaze as an adaptive component of defensive states in humans" has been accepted for publication in Proceedings B.

Data Accessibility section

Open Access

Paper charges

Sincerely,

Appendix A

Dear Dr. Brosnan,

Thank you very much for the positive evaluation, the helpful feedback and the invitation to submit a revised version of our manuscript. We have carefully revised the manuscript along the suggestions of the editor as well as both reviewers and included more detail regarding the rationale and experimental design, specifically as compared to the previous study by Rösler & Gamer (2019). Moreover, we toned down the interpretation of our findings in order not to overstate the results. Unfortunately, during the revisions, we detected an error in our analysis scripts that resulted in a shift of the timecourses by 2 seconds (i.e., the cue period) for some of the analyses. As a consequence, we carefully checked all analyses and updated the results section and figures accordingly. These changes do not affect the main results and interpretations but we sincerely apologize for this mistake! Finally, we had to refrain from collecting more participants to achieve a gender-balance in our sample. The current laboratory rules that were implemented to safeguard against COVID-infections (e.g., a requirement to wear masks and to use contact-free eye-tracking setups) did not allow us to reproduce the experiments under the same conditions as before. We mention this shortcoming in the discussion, but we also highlight below that we do not consider this as a serious confound to the generalizability of our findings because previous studies in healthy populations did not reveal significant differences between males and females for the currently examined mechanism.

Since both reviewers asked for the inclusion of details to all sections of the manuscript and specifically suggested to move methodological details from the supplementary materials to the main text, the manuscript length increased. Unfortunately, its word count now exceeds the limits stated on the submission website. We have limited the manuscript to the absolutely necessary information and already moved several aspects to the substantially extended supplementary materials, but we are unable to further shorten the manuscript while still complying with the changes that were suggested by the reviewers. After all, this is an extensive study including a large number of autonomic and oculomotor readouts. We sincerely hope that the extension of the manuscript is acceptable but we are also willing to move additional details to the supplementary materials if necessary.

Please find below a detailed explanation of how we addressed all points that were raised by the editor and the reviewers. We are convinced that the revised manuscript will be of great interest to the broad readership of *Proceedings of the Royal Society B* and we are looking forward to hearing from you.

Sincerely,

Alma Merscher

Associate Editor

Comments to Author:

This is an interesting and multifaceted study that I believe offers important and novel insights. Two reviewers have provided feedback on this article and both agree that it has merit, but both also provide detailed and thoughtful suggestions as to how the reporting of this study can be enhanced, both with respect to the framing and the clarity of the methodological approach. A careful response to their feedback would greatly enhance this article I believe.

Thank you very much for your thoughtful and detailed remarks on our study. We appreciate your help in improving our manuscript and hope to have addressed your critique adequately. Please find our detailed responses below.

In addition to the reviewers' feedback, I also have some concerns, some of which echo the comments made by the reviewers.

First, and as noted by the authors in this article, this study represents an extension of a previously published study by Roesler and Gamer (2019). Both in the Introduction and Methods, the authors must explicitly outline how this new study differs from the previous one and what novel insights this new approach offers i.e. what novel insights might this new study offer to the field?

We understand your concern and described the experimental design of both the previous and the current study in more detail. Moreover, we emphasized on the novel insights that can be gained from this work. The revised paragraphs in the introduction and methods read as follows:

Introduction, Lines 88 ff:

“Roesler and Gamer (2019) were the first to report reduced visual scanning along with bradycardia and increases in skin conductance during the expectation of an avoidable aversive electro-tactile stimulation. Specifically, participants showed less and longer fixations that were closer to the center of the screen when anticipating an evitable shock they could avoid by pressing the space bar as compared to an inevitable or no shock. Corroborating the idea of an action preparatory mechanism, such narrowed overt attention as well as enhanced bradycardia predicted the speed of threat-escapes on a trial-wise basis. These results align with previous findings indicating a narrowed focus of attention (i.e., lower accuracy when responding to peripheral vs. central stimuli in a maze) while participants had to actively flee from a predator chasing them as compared to when they were not actively haunted [1]. However, as previous research suggests that eye movements follow task-specific predictions of expected events [2], it remains unclear whether these findings relied on the specific experimental situation that required a narrow attentional focus to prepare for the upcoming flight response.

We thus conducted two experiments to investigate whether oculomotor changes upon threat can also be observed in situations requiring a wider focus of attention and whether they

are threat-specific. Using adapted versions of said paradigm by Rösler and Gamer (2019), we assessed gaze dynamics and autonomic responses during the expectation of an avoidable, no or inevitable threat (Experiment 1) or an achievable, no or guaranteed reward (Experiment 2). In Experiment 1, we modified the task to avert the threat such that it required more distributed spatial attention. Instead of a simple button press, subjects could now avoid the aversive electro-tactile stimulation by quickly reacting upon peripherally presented response prompts. In Experiment 2, we transferred the paradigm into a new context to test whether the concomitant inhibition of oculomotor behavior and cardiac output, resembling the previously described freezing-associated bradycardic defense state, represents a threat-specific phenomenon: Instead of shocks, participants could achieve a guaranteed, no or an achievable financial reward that could be won quickly responding to peripherally presented treasure chests.”

Method, Lines 142 ff:

“Based on Rösler and Gamer (2019), participants were presented with a screen depicting naturalistic pictures against a grey background during which eye movements, pupil width, heart rate (HR) and skin conductance (SC) were measured. Experiment 1 included a threat context where participants could receive an individually calibrated, aversive electro-tactile stimulation using a Digitimer Constant Current Stimulator DS7A (Hertfordshire, United Kingdom; see electronic supplementary material) while Experiment 2 involved a reward context where participants could earn money (fixed amount of 0.10 € per trial).

For visual stimulation, we used 60 affectively neutral images depicting naturalistic scenes (768 × 576 pixels, visual angle of 24.00° × 18.11° at a viewing distance of 50 cm) from the McGill Calibrated Colour Image Database [3] presented in a random order. Although the pictures did not include particularly salient features, half of them (randomly determined within each participant) were horizontally flipped to prevent biases in eye movements provoked by an incidental imbalance in visually stimulating features on one side. The experiment was programmed with Presentation® (Neurobehavioral Systems Inc., Version 18.1) and run on a 24” Asus VG248QE display (53.126 × 29.889 cm, 1920 × 1080 pixels, refresh rate 60 Hz). Naturalistic images (8 s) were preceded by a white fixation cross (6.5-8.5 s) turning red, green or yellow (2 seconds). Depending on the color, participants were instructed whether to expect an inevitable shock or no reward (red), no shock or guaranteed reward (green) or an avoidable shock or achievable reward (yellow) after disappearance of the naturalistic picture. Afterwards, the screen turned blank (1 s) in no shock and shock or no reward or reward trials while the shock and reward trials were accompanied by a shock or a reward delivery. Differing from Rösler and Gamer (2019) who used a verbal prompt to press the space bar as fast as possible, an open and a closed door (Experiment 1) or treasure chest (Experiment 2) appeared on either side of the screen in avoidable shock/achievable reward trials (225 × 388 pixels, distance between the center of the screen and the center of the object in visual angle: 19.88°). The position of the open door or open treasure chest was counterbalanced within each participant. Participants were told they could avoid the potential stimulation or gain the reward by a quick joystick movement into the direction of the open door or treasure chest (for an illustration of the experimental design see Fig. 1). In order to ensure that participants would

receive a shock or win a reward in approximately 50% of the trials, they had to be faster than 600 ms in the first five trials. Afterwards, the threshold was individually adjusted to the median of the first five reaction times. Both experiments comprised 60 trials with 20 in each condition.“

Second, while a reasonable number of participants were included in both studies (N=50), 80% of the participants were female. What is the reason for this skew? Ideally additional male participants would be tested. At the very least, the authors must acknowledge and discuss this potential limitation in the generalizability of their data set. Related to this, it would be helpful if the authors provided greater detail about the people they tested i.e. their familiarity with such testing protocols (e.g., were they psychology students) and if any were colorblind which may have influenced their perception of the stimuli.

Although we did not formally preregister the study, we made an a priori decision on collecting a convenience sample comprising 50 participants for each experiment without determining any further restrictive criteria like gender balance. There are previous studies reporting similar heart rate and skin conductance responses during threat of shock in males and females (e.g., [4]–[6]) so that we did not expect any pivotal differences between sexes pertaining to autonomic responding. We are also not aware of gender differences in oculomotor dynamics during the anticipation of an avoidable shock. However, research is scarce on this matter and many previous studies on freezing-reactions only invited female participants (e.g., [7]–[9]) or collected data from similarly imbalanced samples (e.g., [10], [11]). We therefore agree upon the importance of pointing out that the skew in our sample might limit the generalizability of our findings and that future studies with larger and balanced samples should be conducted. We referred to this limitation in the discussion:

Discussion, Lines 431 - 343:

“Finally, although we are not aware of gender differences in autonomic or oculomotor dynamics during the anticipation of avoidable threat in healthy individuals, the gender imbalance in our sample might limit the generalizability of our findings to men.”

Unfortunately, although we considered to expand our sample to include additional male participants, we cannot reproduce the experiments under the exact same conditions as before due to Covid restrictions and changes in our laboratory set-up. Specifically, we are currently not allowed to test participants without wearing facemasks and we had to re-configure our eye-tracking system to allow for contact-free measurements (so-called desktop-mount) as compared to the original data collection where we used the tower-mount configuration of the eye-tracking system. Due to these constraints, we decided not to examine additional participants since these data would presumably not be comparable to the originally acquired data.

Participants in general were mostly University students with only some of them majoring in Psychology (Experiment 1: $N = 12$, Experiment 2: $N = 16$). We thus assume that previous experience with psychological testing protocols was relatively low. We did not ask for color-blindness but since no participant reported difficulties in discerning the cue-colors in our

experiments and behavioral errors were very rare, we do not expect this to have caused any issues regarding the recognition of the three trial-types based on the colors of the cues.

Third, the description of the methods lacked key detail and could not be replicated as described. Specifically related to the testing set up. For example, where the participants tested alone in a room? were other people present? what were the lighting conditions? was it the same room for all participants? Such factors will likely influence the response of the participants and also the accuracy of the eye tracking measures. Such information can be provided as supplementary materials, but must be included.

We agree and now report more detail on the experimental set-up in the supplementary materials:

Supplemental Methods, Lines 6 ff:

“After providing written informed consent, participants read instructions about which color would signal each of the three different outcomes. They were told that the naturalistic pictures appearing after the color cues are irrelevant for the task but could be freely explored. Participants were additionally instructed to fixate the center of the screen when a fixation cross was visible. Subsequently, electrodermal and electrocardiographic electrodes were attached. The electrode for shock delivery was attached to the left calf. Participants were then seated in a dimly lit sound-proof cabin with their head stabilized by a chin rest and forehead bar to prevent confounding movements. With their dominant hand, they reached for the grip of the joystick, which was fixed on the table with lashing straps at a distance where participants could easily move it towards the left and right side. On the horizontal axis (left/right), the placement of the joystick was set in line with the center of the screen. The experimenter sat right outside the cabin during the experiment to monitor its progression. After participants were seated, they underwent a shock calibration procedure. In line with our previous work [12], they were asked to rate the shock intensity on a visual analogue scale from 0 to 10 (VAS; 0 = “Not painful at all” to 10 = “unbearably painful”) with 4 marking the point where the perceived intensity turns from very unpleasant to slightly painful. Shock intensity was incremented from 0 mA in steps of 0.1 mA until reaching a rating of 4. This procedure was repeated three times. The amperages of every repetition that were rated at a VAS level of 4 were then averaged and the result amplified by 50% to counteract habituation effects during the experiment.”

Lastly, and as noted by the reviewers, the analytical approach should be presented in the body of the article. How you analyze your data is key to understand for the interpretation of the results.

Due to length restrictions, we kept this section very brief and moved parts to the supplementary materials but we agree that this information is crucial to understand the interpretation of the results. We therefore included more details on the analytical approach in our methods section:

Method, Lines 235 ff:

“Data preprocessing and analyses were performed using R (Version 3.3.3, Core Team, 2018) on a significance level of 5%. For all dependent variables, we calculated repeated-measures analyses of variance (rmANOVA) with trial type (Experiment 1: inevitable shock, no shock, avoidable shock; Experiment 2: guaranteed reward, no reward, achievable reward) and second (10 or 20 bins including the cue period for skin conductance, heart rate and pupil width and 8 bins restricted to the picture presentation phase for all three metrics of visual exploration) as within-subject factors for each study. Degrees of freedom were adjusted according to the Greenhouse-Geisser method to compensate for potential violations of the sphericity assumption. To specifically compare the respective active condition (Experiment 1: avoidable shock; Experiment 2: achievable reward) with the remaining two conditions, post-hoc t-tests were performed using false discovery rate correction (FDR, [13]) to adjust for alpha-error accumulation (comparisons between the two passive conditions in each experiment are reported in Figures S3-S5).

We additionally computed a generalized linear mixed model (GLMM) for each study using both heart rate and centralization of gaze during the second half of the picture viewing period separately as predictors for reaction times in the avoidable shock and achievable reward trials, respectively. Subject ID was added as a random intercept into the GLMM (for further details on the calculation as well as the internal consistency of these measures see supplementary methods and Table S9).“

Reviewer 1

This manuscript reports an interesting study with comprehensive assessment of responding to certain and uncertain threats and rewards. The paper has many strengths, notably the multiple measures of defensive responding (eye tracking, pupillometry, heart rate, and skin conductance measures), the authors' skill in using these methods, and their excellent data visualization. Also, the phenomenon of gaze centrality and its use as a measure of freezing are intriguing, and the authors replicate some of their group's previous findings regarding oculomotor threat responding, and extend this work by including reward in addition to threat. However, the paper also has some limitations. My largest concern is that the authors do not adequately frame the study in relation to senior author's prior work (Roesler and Gamer, 2019). This study seems to be a replication and extension of this prior study, but the introduction does not provide an adequate summary of the prior study or how specifically the present study builds upon it. Also, given that the study is framed around gaze centrality as a freezing behavior, it would be helpful if the authors discussed the function and perhaps components of scanning behavior and visual exploration, as well as their relation to defensive responding, in more depth in the introduction and discussion. This is particularly important, because it is not entirely clear that centralized gaze in the present study reflects a generalized form of defensive responding, rather than the demands of the present task. Finally, it is unclear why the threat and reward conditions are not included in the same experiment, but instead are completed as two separate experiments.

We appreciate your encouraging response to our study and understand the necessity of commenting on your concerns. Please, find our detailed responses below.

The authors mention a previous study in this line of work (Roesler and Gamer, 2019), but they do not explain how the present study builds incrementally on this previous study, which is surprising, because they are fairly closely replicating this study in experiment 1, and then extending it by examining the effects in relation to reward in experiment 2. The introduction should have a paragraph explaining this previous study—what they did, what they found—and then explain how the present study builds incrementally upon this work. Also, in the discussion, it would be helpful if they spent more time considering the divergence between the studies. Are there some potential reasons why the other oculomotor findings (regarding number of fixations, fixation duration) from Roesler and Gamer (2019) not replicate?

We fully understand this short-coming and significantly extended the current manuscript to explain both the design and results of the previous study and how the current one builds on it.

Introduction, Lines 88 ff:

“Roesler and Gamer (2019) were the first to report reduced visual scanning along with bradycardia and increases in skin conductance during the expectation of an avoidable aversive

electrotactile stimulation. Specifically, participants showed less and longer fixations that were closer to the center of the screen when anticipating an evitable shock they could avoid by pressing the space bar as compared to an inevitable or no shock. Corroborating the idea of an action preparatory mechanism, such narrowed overt attention as well as enhanced bradycardia predicted the speed of threat-escapes on a trial-wise basis. These results align with previous findings indicating a narrowed focus of attention (i.e., lower accuracy when responding to peripheral vs. central stimuli in a maze) while participants had to actively flee from a predator chasing them as compared to when they were not actively haunted [1]. However, as previous research suggests that eye movements follow task-specific predictions of expected events [2], it remains unclear whether these findings relied on the specific experimental situation that required a narrow attentional focus to prepare for the upcoming flight response.

We thus conducted two experiments to investigate whether oculomotor changes upon threat can also be observed in situations requiring a wider focus of attention and whether they are threat-specific. Using adapted versions of said paradigm by Rösler and Gamer (2019), we assessed gaze dynamics and autonomic responses during the expectation of an avoidable, no or inevitable threat (Experiment 1) or an achievable, no or guaranteed reward (Experiment 2). In Experiment 1, we modified the task to avert the threat such that it required more distributed spatial attention. Instead of a simple button press, subjects could now avoid the aversive electrotactile stimulation by quickly reacting upon peripherally presented response prompts. In Experiment 2, we transferred the paradigm into a new context to test whether the concomitant inhibition of oculomotor behavior and cardiac output, resembling the previously described freezing-associated bradycardic defense state, represents a threat-specific phenomenon: Instead of shocks, participants could achieve a guaranteed, no or an achievable financial reward that could be won quickly responding to peripherally presented treasure chests.”

Method, Lines 142 ff:

“Based on Rösler and Gamer (2019), participants were presented with a screen depicting naturalistic pictures against a grey background during which eye movements, pupil width, heart rate (HR) and skin conductance (SC) were measured. Experiment 1 included a threat context where participants could receive an individually calibrated, aversive electrotactile stimulation using a Digitimer Constant Current Stimulator DS7A (Hertfordshire, United Kingdom; see electronic supplementary material) while Experiment 2 involved a reward context where participants could earn money (fixed amount of 0.10 € per trial).

For visual stimulation, we used 60 affectively neutral images depicting naturalistic scenes (768 × 576 pixels, visual angle of 24.00° × 18.11° at a viewing distance of 50 cm) from the McGill Calibrated Colour Image Database [3] presented in a random order. Although the pictures did not include particularly salient features, half of them (randomly determined within each participant) were horizontally flipped to prevent biases in eye movements provoked by an incidental imbalance in visually stimulating features on one side. The experiment was programmed with Presentation® (Neurobehavioral Systems Inc., Version 18.1) and run on a 24” Asus VG248QE display (53.126 × 29.889 cm, 1920 × 1080 pixels, refresh rate 60 Hz). Naturalistic images (8 s) were preceded by a white fixation cross (6.5-8.5 s) turning red, green

or yellow (2 seconds). Depending on the color, participants were instructed whether to expect an inevitable shock or no reward (red), no shock or guaranteed reward (green) or an avoidable shock or achievable reward (yellow) after disappearance of the naturalistic picture. Afterwards, the screen turned blank (1 s) in no shock and shock or no reward or reward trials while the shock and reward trials were accompanied by a shock or a reward delivery. Differing from Rösler and Gamer (2019) who used a verbal prompt to press the space bar as fast as possible, an open and a closed door (Experiment 1) or treasure chest (Experiment 2) appeared on either side of the screen in avoidable shock/achievable reward trials (225 × 388 pixels, distance between the center of the screen and the center of the object in visual angle: 19.88°). The position of the open door or open treasure chest was counterbalanced within each participant. Participants were told they could avoid the potential stimulation or gain the reward by a quick joystick movement into the direction of the open door or treasure chest (for an illustration of the experimental design see Fig. 1). In order to ensure that participants would receive a shock or win a reward in approximately 50% of the trials, they had to be faster than 600 ms in the first five trials. Afterwards, the threshold was individually adjusted to the median of the first five reaction times. Both experiments comprised 60 trials with 20 in each condition. “

We do not have a specific hypothesis on why the results of Rösler and Gamer (2019) regarding fixation numbers and durations did not replicate. However, it is important to note that our previous work was the first study that tested oculomotor dynamics during the anticipation of avoidable threat. It thus seems possible that some effects replicate while others do not and it was also the aim of the current work to examine the robustness of our previously reported findings. To our opinion, the failure to replicate our results regarding fixation numbers and durations could either be related to changes in the experimental setup (please note that correct responding in the current study required a wider focus of attention – see also our response to the next point) or reflect an increased variability of these fixation metrics across trials and participants. Importantly, effect sizes for fixation numbers and durations were also smaller than for center bias estimates in Rösler and Gamer (2019). Moreover, fixation duration measures tended to be less reliable than center bias estimates in the current study (see Table S9). Thus, although we cannot rule out that the currently observed discrepancy to our previous study relies on changes in the experimental setup, our results clearly indicate that a decreased distance of fixations from the center of the screen (i.e., center bias) seems to be a reliable indicator of freezing-like gaze responses in humans. We briefly elaborate on these issues in the revised discussion section:

Discussion, Lines 374 - 378:

“Supporting our first hypothesis, we replicated previously found reductions in visual scanning – denoted by decreasing average fixation distances from the center of the screen (i.e., increased center bias) – when participants expected an avoidable aversive stimulation (vs. inevitable or no stimulation) even though threat-escape required more distributed spatial attention [12] “

Lines 422 - 429:

“In the current study, fixation durations and numbers did just marginally differ between trial types and thereby failed to replicate previous findings [12]. Whether this was due to changes in the experimental design, low reliability of corresponding measurements or genuinely absent effects on these fixation metrics remains to be an important question for future research. In general, centralization of gaze may best reflect oculomotor changes during the expectation of avoidable threat and future studies should explore whether these reductions in visual exploration correlate with reduced bodily movement [14].”

Also, why was the task of Roesler and Gamer (2019) changed, in regards to the behavior that averted shock or obtained reward? In the prior task, text appeared on the screen, and participants had to press spacebar to avoid the shock; in the present task, doors or treasure chests are presented on the edges of the screen (left and right), and one of them is open—participants then move a joystick rapidly towards the open one. On my reading, this change in the task makes the gaze centrality seem more like a product of the task itself, rather than defensive responding (e.g., freezing) more generally, because fixating centrally is probably ideal for responding quickly to the doors or treasure chests presented in the periphery. Avoiding the shock requires identifying which side of the screen the open door is on, and then moving the joystick towards it. If I were a participant in this study, I would want to avoid the shock, and so on trials on which I had an opportunity to avert a shock, I would be focused on preparing for the quick response needed to avert the shock. I would likely prepare by looking at the center of the screen, which would evenly prepare me for responding to an open door on the left or right. Is this an evolved freezing response, or is it just preparing to avert a shock in the specific task situation that the authors have created? Indeed, the finding that centralized gaze predicts the speed of subsequent responding is consistent with this interpretation. I find the prior task with the space bar response better for demonstrating gaze centrality as a more generalized freezing response (i.e., not a task specific response), because the shock-averting response does not require shifting visuospatial attention, and thus there would be less need to prepare one’s gaze (by fixating centrally). Still, I am not totally convinced that increasing centralized gaze prior to an upcoming visual stimulus that cues the possibility for averting shock is a feature of defensive responding that would generalize to other contexts (rather than a behavior specific to the demands of this task). This is one reason why it would be helpful to spend more time discussing visual exploration in the introduction.

Our aim was to create a new task that requires a wider focus of attention than the previous one (pressing the space bar). To do so, we presented two doors at each side of the display, one of which indicated the movement direction to successfully escape from the shock. Although we agree that one strategy to prepare avoiding the shock might be to look at the center of the screen, one could equally argue that looking at one side of the screen is as strategic, as the participant would instantly know whether to move the joystick away from that side or towards it. We would also argue that if fixating at the center would be an optimal strategy to quickly respond to peripherally presented targets, we should observe a similar pattern in

Experiment 2 involving the possibility to earn rewards. However, we observed a very different scanning pattern here that most likely reflected alternating gazes towards the left and right side of the screen during the anticipation phase (see Figure 3b). Finally, response times were predicted by the strength of gaze centralization in a threat context (Experiment 1) but not in the reward context (Experiment 2). To our opinion, and integrating the results of our previous work (Roesler & Gamer, 2019), these findings collectively indicate that the currently observed centralization of gaze is not related to specific experimental demands but rather reflects a robust freezing-like phenomenon during anticipation of threat.

To clarify the reasoning for the adjusted experimental design and to further elaborate on the phenomenon of gaze centralization, we expanded the introduction and discussion section:

Introduction, Lines 88 ff:

“Roesler and Gamer (2019) were the first to report reduced visual scanning along with bradycardia and increases in skin conductance during the expectation of an avoidable aversive electrotactile stimulation. Specifically, participants showed less and longer fixations that were closer to the center of the screen when anticipating an evitable shock they could avoid by pressing the space bar as compared to an inevitable or no shock. Corroborating the idea of an action preparatory mechanism, such narrowed overt attention as well as enhanced bradycardia predicted the speed of threat-escapes on a trial-wise basis. These results align with previous findings indicating a narrowed focus of attention (i.e., lower accuracy when responding to peripheral vs. central stimuli in a maze) while participants had to actively flee from a predator chasing them as compared to when they were not actively haunted [1]. However, as previous research suggests that eye movements follow task-specific predictions of expected events [2], it remains unclear whether these findings relied on the specific experimental situation that required a narrow attentional focus to prepare for the upcoming flight response.”

Discussion, Lines 413 ff:

“Importantly, the unique occurrence of globally reduced visual scanning (i.e., on both the horizontal and the vertical axis) in a threatening context, that did not merely reflect task-specific demands [2] and predicted the speed of avoidance response, highlights the adaptive and threat-specific nature of this oculomotor component of the anticipatory defensive state. Similar to reductions in body sway, decrease of oculomotor activity can be characterized as inhibition of motion. Whether this is caused by lower or higher activity of antagonistic somatomotor or oculomotor muscles, and how this is achieved on the neuronal circuit level remains an important question for future research. Nonetheless, similar temporal dynamics of the reduction in motor activity allow an initial classification of these responses to reflect freezing-like behavioral states that promote fast subsequent defensive actions [10], [15]. In the current study, fixation durations and numbers did just marginally differ between trial types and thereby failed to replicate previous findings [12]. Whether this was due to changes in the experimental design, low reliability of corresponding measurements or genuinely absent effects on these fixation metrics remains to be an important question for future research. In general, centralization of gaze may best reflect oculomotor changes during the expectation of avoidable

threat and future studies should explore whether these reductions in visual exploration correlate with reduced bodily movement [14]. Moreover, it remains unclear whether and how reduced visual scanning interacts with gaze preferences evoked by more heterogeneous or dynamic visual material such as social scenes, video clips or three-dimensional virtual environments (e.g., [16]–[18]).”

Also, another possible interpretation of the results is that participants are more motivated to avoid an uncertain shock than they are to obtain an uncertain reward (hence more gaze preparation for avoidance behavior), and this likely does not reflect differences in the nature of defensive and appetitive responding, but rather absolute valence of the incentives (the shock is probably more unpleasant than the reward is pleasant).

Thank you very much for raising this important point! We acknowledge that we cannot rule out that the intensity of unpleasantness in Experiment 1 was different from the intensity of pleasantness in Experiment 2 and have thus decided to refer to this limitation in the discussion. There have been efforts to account for pain avoidance and reward preference in previous experiments [19] but creating exact comparability between absolute valence of incentives for shocks and rewards is notoriously difficult to accomplish and unattainable post hoc.

However, we are convinced that the conditions were relatively well-matched in the current study because of two reasons: 1) Autonomic data suggests a comparable activation of the sympathetic and parasympathetic branches of the autonomic nervous system in avoidable shock and achievable reward trial types in both experiments (see Figure 4). 2) Behavioral responses were very similar between avoidable shock and achievable reward trials for accuracy (Experiment 1: $M = 91.7\%$, $SD = 9.7\%$; Experiment 2: $M = 89.0\%$, $SD = 9.3\%$; $t_{(97.82)} = 1.44$, $p = .154$) as well as response times (Experiment 1: $M = 489.8$ ms, $SD = 61.2$ ms; Experiment 2: $M = 472.7$ ms, $SD = 56.4$ ms; $t_{(97.36)} = 1.46$, $p = .148$). Collectively, these data indicate that there were no gross differences in the motivation to escape a shock and obtain a reward in the current study. We refer to these issues in the discussion section and included the statistics regarding the behavioral data in the supplementary material.

Discussion, Lines 405 – 412:

“However, the current findings also indicate a shift towards sympathetic activity in the threat context. In this regard, it seems important to note that we did not explicitly match the negative valence of aversive shocks with the positive valence of financial rewards. Ensuring such comparability is extremely difficult [20] but should be a matter of future research. Nevertheless, we believe that the motivational value of negative outcomes in Experiment 1 and positive outcomes in Experiment 2 was somehow comparable given that autonomic as well as behavioral responses were very similar between avoidable shock and achievable reward trials in the current study.”

The paragraph that begins on line 69 is very long. It could be broken into two or three paragraphs and organized a little better. It might be helpful to begin by explaining how freezing is measured in humans, particularly given that the present paper offers a new potential measure of freezing behavior (reduced sway is briefly alluded to later in the paragraph).

We agree and restructured the paragraph accordingly. Moreover, we explained how freezing is measured in humans earlier on.

Introduction, Lines 59 ff:

“Upon distal yet inevitable threat, humans seem to engage in similar defensive responding denoted by reduced body sway (i.e., freezing), as measured by stabilometric platforms, and a co-activation of sympathetic (e.g., heightened skin conductance) and parasympathetic (e.g., bradycardia) branches of the autonomic nervous system [12], [21], [22]. This integrated defense state has been referred to as attentive immobility/freezing, supposedly preparing the individual for further defensive actions, if eventually the threat becomes imminent or escape options appear [11], [23], [24]. Confusingly, attentive immobility in humans has been discussed as both a vulnerability factor for psychopathologies, e.g. [7], [8], [25] and an adaptive action preparatory mechanism [5], [12], [22]. To reconcile these divergent results, it has been suggested that freezing tends to be associated with clinical constructs when participants cannot escape the threat, while it adaptively facilitates action preparation when subsequent harm can be actively avoided [12]. Experimental context thus seems to constitute a main determinant of the behavioral defense state [26], [27]. While attentive immobility describes a state of heightened vigilance and action preparation when escape might still be an option, a lack of escape routes upon imminent threat elicits a more desperate defense state that has also been termed ‘immobility under attack’ [24].”

It would help to describe the content of the images used in the study, for those not familiar with the database. Were they affectively neutral? Did they depict landscapes and objects?

We did not formally assess the affective quality of the images in the current study, but the following selection criteria were established when compiling the stimulus set: 1) The images should be affectively neutral, 2) they should depict naturalistic scenes rather than objects or animals, 3) they should not include particularly salient features and should be relatively uniform and symmetric. We added these details to the methods section:

Method, Lines 149 - 151:

“For visual stimulation, we used 60 affectively neutral images depicting naturalistic scenes (768 × 576 pixels, visual angle of 24.00° × 18.11° at a viewing distance of 50 cm) from the McGill Calibrated Colour Image Database [3] presented in a random order.”

This sentence could be explained further: “The pictures were presented in a random order with half of them horizontally flipped to prevent systematic horizontal biases in eye movements provoked by basic pictorial features.” What exactly is meant by horizontally flipped? What is the horizontal bias in eye movements? A citation regarding the horizontal bias would help if an appropriate one exists. I am familiar with the tendency to first look left; is this the horizontal bias, or does it refer to a general horizontal tendency in eye movements?

Thank you for pointing this out. We mirrored (horizontally flipped) half of the pictures as we wanted to avoid that some of them incidentally happened to depict the most salient visual information on the left or the right side, respectively. This was just a precaution to prevent visual exploration to be biased towards one side of the stimulus material. We do not think that the most relevant pictorial information in the currently selected photographs tended to be on one side but we wanted to rule out this possibility by randomly flipping half of the images for each participant. We modified the above-mentioned sentence to the following:

Method, Lines 151 ff:

“Although the pictures did not include particularly salient features, half of them (randomly determined within each participant) were horizontally flipped to prevent biases in eye movements provoked by an incidental imbalance in visually stimulating features on one side. The experiment was programmed with Presentation® (Neurobehavioral Systems Inc., Version 18.1) and run on a 24” Asus VG248QE display (53.126 × 29.889 cm, 1920 × 1080 pixels, refresh rate 60 Hz).”

I found this wording confusing: “...participants were instructed whether to expect a certain (red; Exp. 1: shock condition, Exp. 2: no reward condition), no (green; Exp. 1: no shock condition, Exp. 2: reward condition) or an avoidable/achievable (yellow; Exp. 1: potential shock condition; Exp. 2: potential reward condition) aversive electrotactile stimulation (Experiment 1) or reward (Experiment 2) after disappearance of the naturalistic picture.” Also, perhaps I am misunderstanding, but it seems like there was a fixed relation between the color of the cue and the condition. This isn’t too serious of a confound, but why not randomize the color-cue pairing between participants?

There was indeed a fixed relation between the color of the cue and the condition. We tried to meaningfully combine the cue colors with the trial outcome so that, similar to a traffic light, red would signal an aversive/unpleasant outcome (either a shock or no reward), green a non-aversive/pleasant outcome (no shock or reward), and yellow a chance to avoid harm/achieve a reward (potential shock or reward). Randomizing the color-cue pairing would thus destroy this storyline that might have helped participants to remember the cue-color associations. It might also be worth mentioning that participants were explicitly instructed about these associations beforehand and did not have to learn them. Moreover, the period during cue presentation was not of main interest to our investigation as we were mainly interested in the anticipation period afterward. Thus, we do not think that the fixed relation between the color-cue and the condition was problematic for the interpretation of the current findings. In addition to clarifying these

details in the supplementary material, we also changed the wording of the respective paragraph (see the following comment).

Although all the durations are listed in the figure, only some are listed in the text (duration of image, which is important, is not listed in text).

We harmonized this and now report all durations in the figure as well as the text.

Method, Lines 157 ff.:

“Naturalistic images (8 s) were preceded by a white fixation cross (6.5-8.5 s) turning red, green or yellow (2 seconds). Depending on the color, participants were instructed whether to expect an inevitable shock or no reward (red), no shock or guaranteed reward (green) or an avoidable shock or achievable reward (yellow) after disappearance of the naturalistic picture. Afterwards, the screen turned blank (1 s) in no shock and shock or no reward or reward trials while the shock and reward trials were accompanied by a shock or a reward delivery.”

I don't understand why the complicated offline drift correction procedure was used. It would seem much more straightforward (and in line with prior research) to remove trials in which participants were not fixating within some distance (e.g., 1 degree of visual angle) from the fixation cross. The authors used the EyeLink's event detection algorithms. Why not use the fixation events and their coordinates when deciding if a participant did not fixate centrally prior to a trial? Also, the offline drift correction struck me as unusual, too. It's my experience that error is not consistent across the screen. Just because the point of gaze estimation at the center for the screen is off in a certain direction by a certain distance does not mean the same error will occur at all points on the screen. I could be wrong about this, but in my experience validating the calibrations on EyeLink systems, the error is not consistent in this manner. I don't doubt that these procedures were appropriate, but with the information provided, I am left confused, and other readers might feel the same way.

We agree that we should elaborate on these procedures in more detail. The algorithm that was used for offline drift correction has been established in our lab a few years ago and we use it since then because it provided very useful across a variety of different eye-tracking systems and recording conditions. Since this algorithm is adaptive, it takes into account the current recording quality. Thus, it is less conservative than more simple approaches but still allows for the reliable detection of problematic trials. Using a simple alternative to the current algorithm (1° of visual angle around the center of the screen) would result in a very similar selection of trials in the current study (both algorithms agree in more than 90% of trials) but overall, more trials would be excluded (13.0% instead of 8.2% in Experiment 1 and 11.3% instead of 7.9% in Experiment 2). Regarding the offline drift correction, we experienced the error caused by slight head movements to be relatively similarly distributed within the area that we used for stimulation (24° × 18° of the screen's center). To our knowledge, the calibration field is also linearly translated when using online drift correction procedures as implemented within the EyeLink software. Finally, the currently derived eye-tracking metrics are either not affected by the drift

correction at all (fixation numbers and durations) or should only show negligible changes when slightly shifting the reference point (average distance of fixations to the center). Due to length restrictions, we only added some of these details as well as a few references to the methods section, but we are willing to elaborate on these procedures in greater detail if necessary.

Method, Lines 191-193:

“To ensure that participants fixated the center of the screen during the baseline, we used our lab’s established iterative outlier detection algorithm [12], [16].”

This result was difficult to interpret: “A generalized linear mixed model (GLMM) with participant IDs as random intercept (see Supplementary Material for further details) demonstrated that global center bias during the second half of picture viewing predicted faster response times in potential shock trials on a trial-wise basis ($\beta = .03$, $SE = .01$, $t(875.13) = 2.32$, $p = .021$).” It might help to add a data analysis plan prior to results to explain the analyses more.

We agree and significantly expanded the methods section to explain our data analysis plan.

Method, Lines 235 ff:

“Statistical analyses

Data preprocessing and analyses were performed using R (Version 3.3.3, Core Team, 2018) on a significance level of 5%. For all dependent variables, we calculated repeated-measures analyses of variance (rmANOVA) with trial type (Experiment 1: inevitable shock, no shock, avoidable shock; Experiment 2: guaranteed reward, no reward, achievable reward) and second (10 or 20 bins including the cue period for skin conductance, heart rate and pupil width and 8 bins restricted to the picture presentation phase for all three metrics of visual exploration) as within-subject factors for each study. Degrees of freedom were adjusted according to the Greenhouse-Geisser method to compensate for potential violations of the sphericity assumption. To specifically compare the respective active condition (Experiment 1: avoidable shock; Experiment 2: achievable reward) with the remaining two conditions, post-hoc t-tests were performed using false discovery rate correction (FDR, [13]) to adjust for alpha-error accumulation (comparisons between the two passive conditions in each experiment are reported in Figures S3-S5).

We additionally computed a generalized linear mixed model (GLMM) for each study using both heart rate and centralization of gaze during the second half of the picture viewing period separately as predictors for reaction times in the avoidable shock and achievable reward trials, respectively. Subject ID was added as a random intercept into the GLMM (for further details on the calculation as well as the internal consistency of these measures see supplementary methods and Table S9).”

Did the authors examine correlations between the oculomotor responding of interest and the psychophysiological measures of defensive responding? I understand that these measures could have low reliability, making it difficult to observe their true relationship, but many readers would be interested in the coherence between these responses.

Yes, we examined correlations between oculomotor and autonomic measures in both experiments. In order to restrict the number of correlations in these exploratory analyses, we concentrated on the second half of the anticipation phase that was also found to be most relevant in our previous study [12]. As expected, the different fixation measures were correlated but correlations between oculomotor and autonomic measures were generally low and differed between experimental conditions. In general, no consistent pattern emerged across both experiments. We included this plot as supplementary Figure S2.

Threat context

Avoidable shock

Reward context

Achievable reward

Inevitable shock

Guaranteed reward

No shock

No reward

Fig. S2. Pearson correlations between fixation numbers (FixN), fixation durations (FixDur), global center bias (CB), pupil width (PW), heart rate (HR) and skin conductance (SC). Values were averaged across the second half of the anticipation phase in Experiment 1 (Threat context) and Experiment 2 (Reward context).

I did not see a description of the shock in the methods section. It would be helpful to provide more information on this feature of the methodology so that the work can be replicated by others. Perhaps the authors referred readers to the group’s prior study and I missed it?

Shock calibration and delivery were similar to Rösler & Gamer (2019). For sake of brevity, we briefly refer to this in the methods section but also provide a detailed description in the supplementary materials.

Methods, Line 144 - 148:

“Experiment 1 included a threat context where participants could receive an individually calibrated, aversive electro tactile stimulation using a Digitimer Constant Current Stimulator DS7A (Hertfordshire, United Kingdom; see electronic supplementary material) while Experiment 2 involved a reward context where participants could earn money (fixed amount of 0.10 € per trial).”

Supplementary material, Line 18 - 24:

“In line with Rösler and Gamer (2019), they were asked to rate the shock intensity on a visual analogue scale from 0 to 10 (VAS; 0 = “Not painful at all” to 10 = “unbearably painful”) with 4 marking the point where the perceived intensity turns from very unpleasant to slightly painful. Shock intensity was incremented from 0 mA in steps of 0.1 mA until reaching a rating of 4. This procedure was repeated three times. The amperages of every repetition that were rated at a VAS level of 4 were then averaged and the result amplified by 50% to counteract habituation effects during the experiment.”

In the figures, a line denotes regions of significant difference. The authors report using some false discovery correction, but I could not find more details in the manuscript.

We used the method described by Benjamini & Hochberg (1995) and added a reference to this article to the methods section.

Method, Lines 241 ff:

“Degrees of freedom were adjusted according to the Greenhouse-Geisser method to compensate for potential violations of the sphericity assumption. To specifically compare the respective active condition (Experiment 1: avoidable shock; Experiment 2: achievable reward) with the remaining two conditions, post-hoc t-tests were performed using false discovery rate correction (FDR, [13]) to adjust for alpha-error accumulation (comparisons between the two passive conditions in each experiment are reported in Figures S3-S5).”

It is also unclear why the power analysis is for the individual experiments, and not for the critical comparison of the two conditions, which the authors ultimately conduct in their analysis. Why not randomly assign participants to the two conditions in one experiment?

Also, out of my own curiosity, would it be impossible to do the two conditions (threat and reward) within subjects?

This was due to logistical reasons. Given the results of Gamer & Rösler (2019), we wondered first whether the oculomotor effects were due to the experimental task or the aversive context. We therefore developed an experimental protocol that requires a wider focus of attention in order to quickly react to the response cue. After conducting Experiment 1, we deemed it worthwhile to further examine whether and which of the observed effects would specifically occur in a threatening context. Since the experiments were conducted sequentially, we do not want to convey the impression that we focused on comparing the threat and the reward context already in the beginning when planning the studies. However, we believe that both experiments should be discussed and interpreted together which is why we included both studies in this article.

In general, on the basis of our experience with this experimental design, we would refrain from using a within-subjects manipulation of the threat/reward-context as we would anticipate substantial carryover effects.

Importantly, although we did the power analysis for each individual experiment, the statistical power for comparisons between experiments is still .80 when considering simple comparisons between effects (one-sided t-test, medium effect size of $d = 0.5$, $\alpha = .05$).

In the prior study, the authors looked at individual differences in anxiety-related traits. Did the authors repeat these analyses in the present study?

We did indeed look at individual differences in anxiety-related traits but these data were only acquired in Experiment 1 (see Table R1 below) since we did not have clear hypotheses regarding such relationship in the reward context (Experiment 2). Therefore, we abstained from including it into the paper. In fact, we replicated the null-findings regarding the relationship between oculomotor and autonomic measures and anxiety-related traits that we also observed in our previous study [12].

Table R1. Correlation coefficients between the self-report anxiety measures and the main dependent variables in Experiment 1.

	Center Bias	Fixation Duration	Fixation Number	Skin Conductance	Heart Rate	Pupil Width
ASI	-.179	.056	.003	-.125	-.259	.038
STAI	-.018	.046	-.045	-.260	.138	-.101

Note. ASI = Anxiety Sensitivity Index; STAI = State-Trait Anxiety Inventory. Autonomic and oculomotor measures were averaged across all time bins during the second half of the stimulus presentation (bin 5 to 8). None of the observed correlations was statistically significant.

It would be helpful to report the internal consistency of the variables. I know this is difficult because the authors looked at trends across the trial. But for the eye tracking variables, I wonder, were gaze centrality, total fixations, and fixation durations consistent across trials when looking at metrics for the entire trial (rather than time points within the trial)? Reporting reliability will help readers determine if these metrics could be used in individual difference research, which would seem important given the clinical framing in the introduction. See Parsons et al. (2019):

Parsons, S., Kruijt, A. W., & Fox, E. (2019). Psychological science needs a standard practice of reporting the reliability of cognitive-behavioral measurements. *Advances in Methods and Practices in Psychological Science*, 2(4), 378-395.

We agree that some estimate of the reliability of the dependent measures seems useful for future research. We therefore calculated the internal consistency for each oculomotor and autonomic measure across all trials in each condition. Similar to the correlation analyses described above, we focused on the second half of the anticipation phase for these calculations. Interestingly, for the majority of measures, internal consistency is rather high across trials within each condition. We included this table in the supplementary materials (Table S9).

Table S9. Internal consistency for all dependent across all trials per trial type.

	Experiment 1: Threat context			Experiment 2: Reward context		
	Avoidable Shock	Inevitable Shock	No Shock	Achievable Reward	Guaranteed Reward	No Reward
Global Center Bias	.94	.87	.88	.95	.94	.90
Fixation Numbers	.96	.91	.89	.97	.94	.92
Fixation Durations	.95	.86	.77	.97	.91	.89
Skin Conductance	.68	.81	.80	.87	.71	.83
Heart Rate	.61	.62	.65	.73	.15	.41
Pupil width	.92	.92	.93	.95	.93	.94

Note. All dependent variables are averaged across the second half of the anticipation phase.

Referee: 2

Comments to the Author(s)

In this interesting study, Merscher and colleagues tested response patterns of heart rate, skin conductance and ocular measures (pupil dilation, eye movements) as indicators of defensive and/or appetitive responding. In their first experiment, autonomous and ocular responses were investigated, evoked while the experimental subjects anticipated inevitable threat of shock, the absence of threat or a threat that could be averted by active avoidance behaviour. Accordingly, in the second experiment, the shock was replaced with a monetary reward and responses were examined during the anticipation of guaranteed reward, the absence of reward or a reward that could be obtained by active approach behaviour. In either experiment there were 50 human participants.

Cardiac deceleration, skin conductance and pupil dilation increased during the anticipation of threat, which was most pronounced for heart rate and pupil dilation when active behaviour was required to avoid the shock. In the reward context, however, cardiac deceleration, skin conductance and pupil dilation seemed to be only increased during the anticipation of reward, when it required active behaviour to be obtained. The investigation of eye movements further revealed particularly reduced visual exploration during the anticipation of evitable threat, while in anticipation of obtainable rewards, exploration was mostly unchanged. The authors, therefore, conclude that the investigated autonomous responses reflect a state of action preparation for both avoiding threat and approaching reward, while freezing of visual exploration may be an exclusive index of defensive response preparation. Correspondingly, cardiac deceleration predicted faster avoidance/approach behaviour, while freezing of visual exploration exclusively predicted faster threat avoidance.

I think the study has been elegantly conducted, is methodologically sound and, in general, well analysed. The results are very important to a broad audience of basic and translational neuroscientists, as they expand our knowledge of the functional aspects and expressions of defensive and appetitive responding. I therefore believe, that this manuscript will arouse much interest in the scientific community. However, I have a few concerns that need to be addressed before the manuscript may be considered ready for publication.

Many thanks for your positive evaluation and for seeing relevance for a broad scientific audience in our work. We greatly appreciate your thorough assessment of our manuscript and will elaborate on your concerns in more detail below.

Abstract

- a) **The authors mainly focus on the response differences during anticipation of threat, that requires action to be avoided (active threat), vs. the anticipation of reward, that requires action to be obtained (active reward). However, the other**

experimental contexts are of interest as well, helping to link the current data to previous literature in the field (e.g., Löw et al., 2008 and 2015). The authors, therefore, should also include their findings on defensive/appetitive responding during anticipation of inevitable threat/guaranteed reward in the abstract and, based on these findings, outline how responses in the active conditions differed.

We agree and rephrased the abstract accordingly. Specifically, we added more information on our findings regarding defensive/appetitive responding to inevitable threats/guaranteed rewards.

Abstract, Lines 26 - 42:

“Adequate defensive responding is crucial for mental health but scientifically not well understood. Specifically, it seems difficult to dissociate defense and approach states based on autonomic response patterns. We thus explored the robustness and threat-specificity of recently described oculomotor dynamics upon threat in anticipation of either threatening or rewarding stimuli in humans. While visually exploring naturalistic images, participants (50 per experiment) expected an inevitable, no, or avoidable shock (Experiment 1) or a guaranteed, no or achievable reward (Experiment 2) that could be averted or gained by a quick behavioral response. We observed reduced heart rate (bradycardia), increased skin conductance, pupil dilation, and globally centralized gaze when shocks were inevitable but, more pronouncedly, when they were avoidable. Reward trials were not associated with globally narrowed visual exploration, but autonomic responses resembled characteristics of the threat condition. While bradycardia and concomitant sympathetic activation reflect not only threat-related but also action-preparatory states independent of valence, centralization of gaze seems a robust and specific phenomenon during the anticipation of threat. Thus, instead of relying on single readouts, translational research in animals and humans should consider the multi-dimensionality of states in aversive and rewarding contexts, especially when investigating ambivalent, conflicting situations.”

- b) The conditions during which action was required for threat avoidance/reward are termed “potential threat” or “potential reward”, respectively. As far as I’m aware of, the term “potential” has primarily been used in past research to highlight that the occurrence of threat/reward is unpredictable, but inevitable (see Davis et al., 2010 or Schmitz et al., 2011). The authors should therefore rephrase the potential condition either to active threat/active reward (as done in Löw et al., 2015) or evitable threat/obtainable reward to avoid misunderstandings. This would also apply to the entire paper.**

Thank you very much for this remark. We agree that the term “potential” could be misunderstood and correspondingly replaced potential threat/reward with avoidable threat/achievable reward, and certain threat/reward with inevitable threat/guaranteed reward throughout the entire paper.

c) It should be stated, that there were two experiments

This information is now included in the abstract.

- d) Only a minor suggestion: The authors may want to consider to focus less on defensive responding in the abstract, but frame the findings in a predator-prey/defense-approach context, as previously done (see Löw et al., 2008 or Lang et al., 1997) – i.e., that physiological responses in preys preparing for avoidance resemble those in predators preparing for approach, providing an excellent link to the study. This may also help to broaden the scope of the manuscript, highlight the novelty of the findings and enhance the impact in the scientific community. This would also apply for the introduction and the discussion.**

We appreciate your suggestion. However, as we were mainly interested in elucidating the robustness and threat-specificity of oculomotor and autonomic dynamics upon threat and their relevance to anxiety-related disorders [12], we decided to keep defensive responding as our main focus. This also reflects our experimental strategy since we designed the second experiment as a control to test whether and which of the effects we found in the first experiment are threat-specific. However, we still deem your point very relevant and therefore slightly expanded the introduction and discussion section to broaden the scope of the manuscript regarding this aspect.

Introduction, Line 78 ff:

“Heart rate, in contrast, was only associated with faster motor responses. Diverging from a widespread idea that due to its robust co-occurrence, e.g., [14] bradycardia might be adduced as a proxy for freezing e.g., [28], [29], cardiac and motor inhibition seem to reflect different aspects of a defense state. Previous studies using active contexts suggested that transient bradycardia indeed also occurs independently of threat, constituting a more general action-preparatory mechanism [10], [30]–[32]. This goes along with the notion that appetitive and defensive responding bear fundamental similarities in that they require the organism to anticipate and prepare for subsequent actions [31], [33].”

Discussion, Lines 400 - 405:

“The overall pattern of skin conductance changes and pupil dilation in conjunction with heart rate deceleration indicates a co-activation of sympathetic and parasympathetic branches of the autonomic nervous system in threat and reward contexts which suggests some resemblance of physiological responses in prey preparing to avoid harm and predators preparing for approach [31].”

Introduction

- a) My major concern is, that the concept of freezing is not elaborated in enough detail, which made it hard to follow the author's thoughts. Introducing attentive immobility/attentive freezing in the context of the dynamic organization of defensive responding (see Hamm, 2020) may provide a more profound basis for the study's design. It may also help to increase comprehensibility of the introduction. Here, attentive immobility renders a state of vigilant readiness during confrontations with inevitable but rather distal threat, where the organism monitors the source of danger and prepares for defensive action, if the threat becomes more imminent or an avoidance option eventually opens up, which resembles the situation of the participants in the current study (see also Lang, 2000). Cardiac deceleration (so called fear bradycardia) as well as high responsivity to external events are hallmarks of this defensive state, while during more imminent threat stages, when defensive action is required, heart rate supportively accelerates, which resembles the 1-s action period in the current study. Furthermore, attentive immobility may, thus, be separated from tonic immobility, evoked during imminent threat stages and associated with cardiac acceleration and unresponsiveness (see Volchan et al., 2017). The authors may also be interested in further literature on the relationship between heart rate and attentive immobility (Szeska et al., 2021).

Thank you very much for your suggestions and for pointing us to the additional relevant literature. We agree that the concept of freezing was not explained clearly enough and that the differentiation from tonic immobility was somewhat misleading. We rephrased the entire introduction so that it includes the dynamic organization of defensive behaviors in both animals and humans and referred to Volchan et al.'s (2017) framework regarding the integrated defensive state that occurs during the anticipation of inevitable but relatively distant threat.

Introduction, Lines 47 ff:

“Various forms of defensive behaviors have evolved to protect an organism from potential harm in threatening situations [34], [35]. Depending on the context (e.g., availability of escape routes; temporal and spatial distance of a threat), they occur in a cascade-like fashion ranging from hard-wired automatic, initial reactions to deliberative goal-directed behaviors [21], [36]–[39].

An evolutionarily conserved response in the face of real or perceived threat is a defensive behavioral pattern eventually termed freezing [8]. Most of this research was concentrated on rodents and freezing was characterized by movement cessation, accompanied by a transient decrease in heart rate, i.e. bradycardia [41]–[44]. This defensive mode of simultaneous behavioral and cardiovascular inhibition has been suggested to help avoiding predator detection [45] optimizing perceptual and attentional processing [46], [47], and to prepare fast responses to approaching threat[48], [49].

Upon distal yet inevitable threat, humans seem to engage in similar defensive responding denoted by reduced body sway (i.e., freezing), as measured by stabilometric platforms, and a

co-activation of sympathetic (e.g., heightened skin conductance) and parasympathetic (e.g., bradycardia) branches of the autonomic nervous system [12], [21], [22]. This integrated defense state has been referred to as attentive immobility/freezing, supposedly preparing the individual for further defensive actions, if eventually the threat becomes imminent or escape options appear [11], [23], [24].”

b) Based on the previous comment, I disagree with the general assumption of the authors, that reduced body sway in passive contexts – which are reliably associated with cardiac decelerations – reflect tonic immobility (lines 73 and 74)

We fully understand your reasoning and agree with your concerns. In the modified version of our introduction we thus refrained from defining reduced body sway in passive contexts as tonic immobility.

Introduction, Lines 65 ff:

“Confusingly, attentive immobility in humans has been discussed as both a vulnerability factor for psychopathologies, e.g. [7], [8], [25] and an adaptive action preparatory mechanism [5], [12], [22]. To reconcile these divergent results, it has been suggested that freezing tends to be associated with clinical constructs when participants cannot escape the threat, while it adaptively facilitates action preparation when subsequent harm can be actively avoided [12]. Experimental context thus seems to constitute a main determinant of the behavioral defense state [26], [27]. While attentive immobility describes a state of heightened vigilance and action preparation when escape might still be an option, a lack of escape routes upon imminent threat elicits a more desperate defense state that has also been termed ‘immobility under attack’ [24].”

c) The authors may also want to consider implementing previous research by Löw and colleagues (2008), where changes in skin conductance and heart rate were also measured at varying stages of threat/reward imminence, including stages of action preparation for avoidance/approach.

We amended our introduction so that it now also includes findings from Löw and colleagues (2008).

Introduction, Lines 78 ff:

“Heart rate, in contrast, was only associated with faster motor responses. Diverging from a widespread idea that due to its robust co-occurrence, e.g., [14] bradycardia might be adduced as a proxy for freezing e.g., [28], [29], cardiac and motor inhibition seem to reflect different aspects of a defense state. Previous studies using active contexts suggested that transient bradycardia indeed also occurs independently of threat, constituting a more general action-preparatory mechanism [10], [30]–[32]. This goes along with the notion that appetitive and defensive responding bear fundamental similarities in that they require the organism to

anticipate and prepare for subsequent actions [31], [33]. Responses that specifically index and discriminate between threat-induced, defensive and reward-related, appetitive states in humans are thus necessary to understand fear-associated neural circuitries and behaviors.”

Methods

- a) The authors should specify, what was problematic about the eye tracking data or made the heart rate invalid, that justified an exclusion of the respective participants from the analyses. This is important, especially given the current controversy on excluding participants on the basis of response measures (see Lonsdorf et al., 2019)**

We agree that researchers should clearly explain the specific reasons for excluding participants and we therefore added this information to the methods section.

Method, Lines 126 - 131:

“In Experiment 1, seven participants were excluded due to problematic eye tracking data (i.e., more than 30% of eye-tracking trials with baseline outliers or missing baseline position data, or a range of baseline coordinates exceeding 5° of visual angle after trial exclusion) and another one because of frequent extrasystoles in the heart rate data resulting in a total of 50 participants (40 women, age: $M = 28.00$ years, $SD = 10.79$ years).”

- b) The 8-s duration of the presentation of the naturalistic pictures should also be mentioned in the text, not only in Figure 1**

We agree and added this information to the methods section.

Method, Lines 157 ff:

“Naturalistic images (8 s) were preceded by a white fixation cross (6.5-8.5 s) turning red, green or yellow (2 seconds). Depending on the color, participants were instructed whether to expect an inevitable shock or no reward (red), no shock or guaranteed reward (green) or an avoidable shock or achievable reward (yellow) after disappearance of the naturalistic picture. Afterwards, the screen turned blank (1 s) in no shock and shock or no reward or reward trials while the shock and reward trials were accompanied by a shock or a reward delivery.”

- c) I missed sections, where the electrical shock is described, especially whether the intensity was fixed or individually adjusted. If the latter is the case (which I assume, since has been done in the first investigation from 2019), the authors should provide some details on the shock workup at least in the supplemental material.**

Shock calibration and delivery were similar to Rösler and Gamer (2019). We added this information to the methods section and explain the shock calibration procedure in detail in the supplementary material.

Method, Lines 144 ff:

“Experiment 1 included a threat context where participants could receive an individually calibrated, aversive electrotactile stimulation using a Digitimer Constant Current Stimulator DS7A (Hertfordshire, United Kingdom; see electronic supplementary material) while Experiment 2 involved a reward context where participants could earn money (fixed amount of 0.10 € per trial).”

Supplementary material, Line 18 - 24:

“In line with Rösler and Gamer (2019), they were asked to rate the shock intensity on a visual analogue scale from 0 to 10 (VAS; 0 = “Not painful at all” to 10 = “unbearably painful”) with 4 marking the point where the perceived intensity turns from very unpleasant to slightly painful. Shock intensity was incremented from 0 mA in steps of 0.1 mA until reaching a rating of 4. This procedure was repeated three times. The amperages of every repetition that were rated at a VAS level of 4 were then averaged and the result amplified by 50% to counteract habituation effects during the experiment.”

- d) A short sentence at the end of the methods should provide some general details on the statistical analyses and refer to the supplements for further reading.**

Since Reviewer #1 also suggested to include more details on our data analysis plan, we significantly expanded the methods section to provide more information on the statistical analyses.

Methods, Lines 235 ff:

“Data preprocessing and analyses were performed using R (Version 3.3.3, Core Team, 2018) on a significance level of 5%. For all dependent variables, we calculated repeated-measures analyses of variance (rmANOVA) with trial type (Experiment 1: inevitable shock, no shock, avoidable shock; Experiment 2: guaranteed reward, no reward, achievable reward) and second (10 or 20 bins including the cue period for skin conductance, heart rate and pupil width and 8 bins restricted to the picture presentation phase for all three metrics of visual exploration) as within-subject factors for each study. Degrees of freedom were adjusted according to the Greenhouse-Geisser method to compensate for potential violations of the sphericity assumption. To specifically compare the respective active condition (Experiment 1:

avoidable shock; Experiment 2: achievable reward) with the remaining two conditions, post-hoc *t*-tests were performed using false discovery rate correction (FDR, [13]) to adjust for alpha-error accumulation (comparisons between the two passive conditions in each experiment are reported in Figures S3-S5).

We additionally computed a generalized linear mixed model (GLMM) for each study using both heart rate and centralization of gaze during the second half of the picture viewing period separately as predictors for reaction times in the avoidable shock and achievable reward trials, respectively. Subject ID was added as a random intercept into the GLMM (for further details on the calculation as well as the internal consistency of these measures see supplementary methods and Table S9).“

Results

a) Similar to the abstract, my major concern with the results is that the authors primarily focused on distinguishing the active threat/reward condition from the other two. Instead, in my view, it would be better to first characterize how the certain threat/certain reward condition differed from the no threat/no reward condition and then report, how the active conditions evoked different response patterns. In other words: The authors should disentangle the interactions they found in more detail. This may also answer the following questions I had while reading the results section:

We understand that elaborating on how the certain threat/reward conditions differed from the other two conditions might broaden the scope of our paper. We thus included these findings to the results and discussion sections. Moreover, we added Figures S3-S5 to the online supplement including direct contrasts between the inevitable vs. no threat and the guaranteed vs. no reward conditions. Nonetheless, also due to the tight length restrictions of the journal, our central focus remains on the active threat condition that goes along with our main oculomotor finding (centralization of gaze) and most closely resembles the previously found state of attentive immobility preparing the organism for subsequent defensive actions.

Results, Lines 156 ff:

“Oculomotor behavior during the anticipation of threat and reward

Using a 3×8 rmANOVA, we first compared the average distances of fixations from the center of the screen (global center bias) during image presentation between inevitable, no and avoidable shock trials. Visual exploration decreased markedly towards the end of the anticipation period when participants awaited an avorable shock as compared to both other conditions (Fig. 2a; interaction trial type \times second, $F_{(14,686)} = 13.64$, $\varepsilon = .31$, $p < .001$, $\eta^2_g = .03$; main effects for this and the following analyses are reported in Table S1 and Table S2). No robust differences were observed between the inevitable and the no shock condition (see Fig. S3). In avoidable shock trials, global center bias predicted faster response times on a trial-wise basis ($\beta = .04$, $SE = .01$, $t_{(875.24)} = 2.42$, $p = .016$).“

Lines 314 ff:

“Autonomic responses during the anticipation of threat and reward

To compare heart rate changes between conditions, we performed a 3×10 rmANOVA (trial type by seconds, now also including the 2s cue period) for each experiment. In the aversive context (Experiment 1), average heart rate increased right after cue onset, decreased over the anticipation period and increased again after picture offset across all conditions. This dynamic became gradually more pronounced from no to inevitable to avoidable shock trials but statistically significant differences only emerged between the avoidable shock and the other two conditions (Fig. 4a and Fig. S4, interaction trial type \times second, $F_{(18,882)} = 13.07$, $\varepsilon = .27$, $p < .001$, $\eta^2_g = .04$). In the reward context, achievable reward trials but not guaranteed or no reward trials showed similar heart rate trends with a marked increase after cue onset followed by a decrease and another increase after picture offset (Fig. 4d; interaction trial type \times second, $F_{(18,882)} = 23.38$, $\varepsilon = .28$, $p < .001$, $\eta^2_g = .09$).”

- d) Figure 4a seems to indicate, that cardiac deceleration gradually increases from no-threat to certain-threat to avoidable-threat, primarily during seconds 6-8, which circumscribes the time of the D2-component of cardiac responding, suggested to indicate threat anticipation (see Szeska et al., 2021). As, again, the authors only report that cardiac deceleration differed significantly between the active-threat condition and the other two, I was wondering if this impression proves right?**

Unfortunately, as also stated in our response to editor above, we detected an error in our analysis scripts that resulted in a shift of the timecourses by 2 seconds (i.e., the cue period) for some of the analyses. As a consequence, we carefully checked all analyses and updated the results section and figures accordingly. We apologize for this mistake! Regarding your query, we agree that Figure 4A seems to indicate a graded heart rate deceleration at the end of the anticipation phase from no shock via the inevitable shock to the avoidable shock condition. However, formal t-tests with FDR correction do not substantiate this assumption and we only observed a significant difference between the active condition (avoidable shock) to the other two conditions (see also Figure S4). We included this description in the results section and the discussion:

Results, Lines 316 ff:

“In the aversive context (Experiment 1), average heart rate increased right after cue onset, decreased over the anticipation period and increased again after picture offset across all conditions. This dynamic became gradually more pronounced from no to inevitable to avoidable shock trials but statistically significant differences only emerged between the avoidable shock and the other two conditions (Fig. 4a and Fig. S4, interaction trial type \times second, $F_{(18,882)} = 13.07$, $\varepsilon = .27$, $p < .001$, $\eta^2_g = .04$). In the reward context, achievable reward trials but not guaranteed or no reward trials showed similar heart rate trends with a marked increase after cue onset followed by a decrease and another increase after picture offset (Fig. 4d; interaction trial type \times second, $F_{(18,882)} = 23.38$, $\varepsilon = .28$, $p < .001$, $\eta^2_g = .09$).”

- e) **In contrast, Figure 4d seems to indicate, that while cardiac deceleration was particularly pronounced in the active reward condition, the other two reward conditions did not differ. Is that right?**

That is correct. We only observed a significant deceleration in the achievable reward trials (i.e., the active condition) in comparison to both other conditions. This is now also explicitly shown in Figure S4 and briefly mentioned in the results section.

Results, Lines 322:

“In the reward context, achievable reward trials but not guaranteed or no reward trials showed similar heart rate trends with a marked increase after cue onset followed by a decrease and another increase after picture offset (Fig. 4d; interaction trial type \times second, $F_{(18,882)} = 23.38$, $\varepsilon = .28$, $p < .001$, $\eta^2_g = .09$).”

- f) **If b) and c) are true and especially if cardiac deceleration is additionally stronger in certain shock compared to certain reward trials, as might be concluded from the significant three-way-interaction (lines 278-283) and a comparison of Figures 4a and d, this would indicate that cardiac deceleration is indeed more specific/more sensitive to processing of threat relative to processing of reward. In that case, the discussion should be rephrased.**

We agree that we have not been precise regarding this issue in the original version of the manuscript and now included all relevant information concerning potential differences between the threat and the reward condition to the manuscript as well as the supplementary material. To sum up, we do not see evidence for a higher sensitivity of the cardiac deceleration to the processing of threat relative to reward. What is evident, however, is an increased sympathetic activity in terms of skin conductance increase and pupil dilation in the inevitable shock compared to the guaranteed reward condition. It thus seems that both branches of the autonomic system are activated in parallel by the experimental design employed here but that threat results in a shift of this balance towards a higher sympathetic activation. We explicitly mention and discuss this aspect in the revised discussion section.

Discussion, Lines 400 ff:

“The overall pattern of skin conductance changes and pupil dilation in conjunction with heart rate deceleration indicates a co-activation of sympathetic and parasympathetic branches of the autonomic nervous system in threat and reward contexts which suggests some resemblance of physiological responses in prey preparing to avoid harm and predators preparing for approach [31]. However, the current findings also indicate a shift towards sympathetic activity in the threat context. In this regard, it seems important to note that we did not explicitly match the negative valence of aversive shocks with the positive valence of financial rewards. Ensuring such comparability is extremely difficult [20] but should be a matter of future research.

Nevertheless, we believe that the motivational value of negative outcomes in Experiment 1 and positive outcomes in Experiment 2 was somehow comparable given that autonomic as well as behavioral responses were very similar between avoidable shock and achievable reward trials in the current study.”

- g) The authors should at least mention the profound cardiac acceleration at the end of picture viewing in both active conditions, which might mark the transition from attentive immobility to active avoidance/approach. Given that, the cardiac acceleration at the end of the certain threat condition should also be mentioned and discussed later.**

As mentioned above, the cardiac acceleration occurs about 2 seconds later. It is therefore not clear to what degree it indicates a transition from attentive immobility to active avoidance/approach or mainly reflects the outcome of the trial (i.e., shock or reward). However, we decided to describe the whole progression of heart rate starting with cue onset and ending with trial outcome in both the results section and the discussion in more detail.

Results, Lines 315 ff:

“To compare heart rate changes between conditions, we performed a 3×10 rmANOVA (trial type by seconds, now also including the 2s cue period) for each experiment. In the aversive context (Experiment 1), average heart rate increased right after cue onset, decreased over the anticipation period and increased again after picture offset across all conditions. This dynamic became gradually more pronounced from no to inevitable to avoidable shock trials but statistically significant differences only emerged between the avoidable shock and the other two conditions (Fig. 4a and Fig. S4, interaction trial type \times second, $F_{(18,882)} = 13.07$, $\varepsilon = .27$, $p < .001$, $\eta^2_g = .04$). In the reward context, achievable reward trials but not guaranteed or no reward trials showed similar heart rate trends with a marked increase after cue onset followed by a decrease and another increase after picture offset (Fig. 4d; interaction trial type \times second, $F_{(18,882)} = 23.38$, $\varepsilon = .28$, $p < .001$, $\eta^2_g = .09$).”

Discussion

- a) **As for the introduction, the authors should discuss the results also in the context of Löw et al. (2008), who also investigated autonomous measures during threat and reward anticipation.**

We agree and referred to the paper in the discussion as suggested.

Discussion, Lines 396 - 400:

“Thus, contrasting the idea that transient bradycardia might be more sensitive to threat than reward processing [31], heart rate deceleration during anticipation of a response seems to be an important element of a more general action-preparatory mechanism independent of contextual valence which may support processes of attentional orienting and motor preparation [30], [32].”

- b) **Line 339: The authors should be cautious at suggesting, that cardiac deceleration is independent of contextual valence. Löw et al. (2008) found stronger cardiac deceleration during threat anticipation relative to reward anticipation.**

This is a valid remark. However, in the current study, we did not observe such difference between threat and reward anticipation. We correspondingly refer to heart rate deceleration as rather reflecting a more general action-preparatory mechanism (see our answer to a) above). However, we also acknowledge the difficulty in precisely matching the negative valence of electrotactile stimuli to the positive valence of financial rewards in the discussion section.

Discussion, Lines 405 ff:

“However, the current findings also indicate a shift towards sympathetic activity in the threat context. In this regard, it seems important to note that we did not explicitly match the negative valence of aversive shocks with the positive valence of financial rewards. Ensuring such comparability is extremely difficult [20] but should be a matter of future research. Nevertheless, we believe that the motivational value of negative outcomes in Experiment 1 and positive outcomes in Experiment 2 was somehow comparable given that autonomic as well as behavioral responses were very similar between avoidable shock and achievable reward trials in the current study.”

- c) **Do the authors have any explanation for the failed replication (lines 356-358)?**

We do not have a specific hypothesis on why the results of Rösler and Gamer (2019) regarding fixation numbers and durations did not replicate. However, it is important to note that our previous work was the first study that tested oculomotor dynamics during the anticipation of avoidable threat. It thus seems possible that some effects replicate while others do not and it was also the aim of the current work to examine the robustness of our previously reported findings. To our opinion, the failure to replicate our results regarding fixation numbers and

durations could either be related to changes in the experimental setup (please note that correct responding in the current study required a wider focus of attention) or reflect an increased variability of these fixation metrics across trials and participants. Importantly, effect sizes for fixation numbers and durations were also smaller than for center bias estimates in Rösler and Gamer (2019). Moreover, fixation duration measures tended to be less reliable than center bias estimates in the current study (see Table S9). Thus, although we cannot rule out that the currently observed discrepancy to our previous study relies on changes in the experimental setup, our results clearly indicate that a decreased distance of fixations from the center of the screen (i.e., center bias) seems to be a reliable indicator of freezing-like gaze responses in humans. We briefly elaborate on these issues in the revised discussion section:

Discussion, Lines 374 - 378:

“Supporting our first hypothesis, we replicated previously found reductions in visual scanning – denoted by decreasing average fixation distances from the center of the screen (i.e., increased center bias) – when participants expected an avoidable aversive stimulation (vs. inevitable or no stimulation) even though threat-escape required more distributed spatial attention [12] “

Lines 422 ff:

“In the current study, fixation durations and numbers did just marginally differ between trial types and thereby failed to replicate previous findings [12]. Whether this was due to changes in the experimental design, low reliability of corresponding measurements or genuinely absent effects on these fixation metrics remains to be an important question for future research. In general, centralization of gaze may best reflect oculomotor changes during the expectation of avoidable threat and future studies should explore whether these reductions in visual exploration correlate with reduced bodily movement [14].”

d) Given the results from Szeska et al. (2021), the authors may want to tone down the conclusion drawn in lines 393-396

We appreciate your suggestion and toned down our conclusion accordingly. Heart rate deceleration may initially be elicited by motivationally significant cues indexing heightened orienting and perceptual processing regardless of contextual valence while prolonged bradycardia might differentiate between inevitable threat and safety contexts [11]. However, while our data descriptively shows that bradycardia gradually increases from no shock to inevitable shock to avoidable shock, we failed to find significant differences between no shock and inevitable shock trials. Thus, our data seem to differ from Szeska et al. (2021) regarding this matter but it should also be mentioned that a different experimental design was used here. We decided to mention previous work on cardiovascular responses in threatening contexts in our discussion, but we also highlight the differences that were observed in the current study. Moreover, we adjusted our interpretation to focus on contexts that require an active response, since such situations are most interesting regarding the current study and elicited the most pronounced centralization of gaze.

Discussion, Lines 392 ff:

“Although the observed heart rate pattern consisting of an initial increase followed by a decrease and another increase starting shortly before trial outcome in Experiment 1 is consistent with the so-called cardiac defense [50] and lines up with previous findings on specific fear states in both rodents [51] and humans [5], [11], [18], [49], but see [10], it was also evident on achievable reward trials in Experiment 2. Thus, contrasting the idea that transient bradycardia might be more sensitive to threat than reward processing [31], heart rate deceleration during anticipation of a motor response seems to be an important element of a more general action-preparatory mechanism independent of contextual valence which may support processes of attentional orienting and motor preparation [30], [32].”

Lines 437 ff:

“In contrast, bradycardia alone seems to be less threat-specific but instead reflects a more general action-preparatory mechanism when a motor response is required (regarding inevitable threat see [11]). These findings confirm that defensive responding, which integrates behavioral and autonomic functions (among other components, such as endocrine responses) bear complex temporal dynamics that need to be considered when using them as indicators of fear. Instead of relying on single output measures, integrated analyses of multiple readouts appear more appropriate to define such defense states, which seems relevant to both animal and human research. Future studies should hence consider the multifaceted nature of heart rate deceleration when adducing it as a proxy for fear-related behavior (e.g., in neuroimaging environments, [29]), particularly when creating ambivalent experimental conditions involving conflicts between threat and reward possibilities. “

References

- [1] K. Vaughn and D. Brasier, “The Effects of Fear States from Passive and Active Threats on Breadth of Attention,” p. 10, 2018.
- [2] J. Anderson, H. B. Barlow, R. L. Gregory, M. F. Land, and S. Furneaux, “The knowledge base of the oculomotor system,” *Philos. Trans. R. Soc. Lond. B. Biol. Sci.*, vol. 352, no. 1358, pp. 1231–1239, Aug. 1997, doi: 10.1098/rstb.1997.0105.
- [3] A. Olmos and F. A. A. Kingdom, “A Biologically Inspired Algorithm for the Recovery of Shading and Reflectance Images,” *Perception*, vol. 33, no. 12, pp. 1463–1473, Dec. 2004, doi: 10.1068/p5321.
- [4] M. M. Bradley, T. Silakowski, and P. J. Lang, “Fear of pain and defensive activation,” *PAIN®*, vol. 137, no. 1, pp. 156–163, Jun. 2008, doi: 10.1016/j.pain.2007.08.027.
- [5] M. M. Hashemi *et al.*, “Human defensive freezing: Associations with hair cortisol and trait anxiety,” *Psychoneuroendocrinology*, vol. 133, p. 105417, Nov. 2021, doi: 10.1016/j.psyneuen.2021.105417.
- [6] L. Clark *et al.*, “Risk-avoidant decision making increased by threat of electric shock,” *Psychophysiology*, vol. 49, no. 10, pp. 1436–1443, 2012, doi: 10.1111/j.1469-8986.2012.01454.x.
- [7] K. Roelofs, M. A. Hagens, and J. Stins, “Facing Freeze: Social Threat Induces Bodily Freeze in Humans,” *Psychol. Sci.*, vol. 21, no. 11, pp. 1575–1581, Nov. 2010, doi: 10.1177/0956797610384746.
- [8] M. A. Hagens, J. F. Stins, and K. Roelofs, “Aversive life events enhance human freezing responses,” *J. Exp. Psychol. Gen.*, vol. 141, no. 1, pp. 98–105, 2012, doi: 10.1037/a0024211.
- [9] E. M. Drabant *et al.*, “Experiential, autonomic, and neural responses during threat anticipation vary as a function of threat intensity and neuroticism,” *NeuroImage*, vol. 55, no. 1, pp. 401–410, Mar. 2011, doi: 10.1016/j.neuroimage.2010.11.040.
- [10] A. Löw, M. Weymar, and A. O. Hamm, “When Threat Is Near, Get Out of Here: Dynamics of Defensive Behavior During Freezing and Active Avoidance,” *Psychol. Sci.*, vol. 26, no. 11, pp. 1706–1716, Nov. 2015, doi: 10.1177/0956797615597332.
- [11] “Attentive immobility in the face of inevitable distal threat—Startle potentiation and fear bradycardia as an index of emotion and attention - Szeska - 2021 - Psychophysiology - Wiley Online Library.” <https://onlinelibrary.wiley.com/doi/full/10.1111/psyp.13812> (accessed Feb. 28, 2022).
- [12] L. Rösler and M. Gamer, “Freezing of gaze during action preparation under threat imminence,” *Sci. Rep.*, vol. 9, no. 1, p. 17215, Dec. 2019, doi: 10.1038/s41598-019-53683-4.
- [13] Y. Benjamini and Y. Hochberg, “Controlling the False Discovery Rate: A Practical and Powerful Approach to Multiple Testing,” *J. R. Stat. Soc. Ser. B Methodol.*, vol. 57, no. 1, pp. 289–300, 1995, doi: 10.1111/j.2517-6161.1995.tb02031.x.
- [14] K. Roelofs, “Freeze for action: neurobiological mechanisms in animal and human freezing,” *Philos. Trans. R. Soc. B Biol. Sci.*, vol. 372, no. 1718, p. 20160206, Apr. 2017, doi: 10.1098/rstb.2016.0206.
- [15] T. E. Gladwin, M. M. Hashemi, V. van Ast, and K. Roelofs, “Ready and waiting: Freezing as active action preparation under threat,” *Neurosci. Lett.*, vol. 619, pp. 182–188, Apr. 2016, doi: 10.1016/j.neulet.2016.03.027.
- [16] A. End and M. Gamer, “Preferential Processing of Social Features and Their Interplay with Physical Saliency in Complex Naturalistic Scenes,” *Front. Psychol.*, vol. 8, Mar. 2017, doi: 10.3389/fpsyg.2017.00418.
- [17] “Individual differences in visual salience vary along semantic dimensions,” *PNAS*. <https://www.pnas.org/doi/abs/10.1073/pnas.1820553116> (accessed Feb. 28, 2022).

- [18] M. Rubo and M. Gamer, "Social content and emotional valence modulate gaze fixations in dynamic scenes," *Sci. Rep.*, vol. 8, no. 1, Art. no. 1, Feb. 2018, doi: 10.1038/s41598-018-22127-w.
- [19] S. Qi, D. Hassabis, J. Sun, F. Guo, N. Daw, and D. Mobbs, "How cognitive and reactive fear circuits optimize escape decisions in humans," *Proc. Natl. Acad. Sci.*, vol. 115, no. 12, pp. 3186–3191, Mar. 2018, doi: 10.1073/pnas.1712314115.
- [20] M. Andreatta and P. Pauli, "Appetitive vs. Aversive conditioning in humans," *Front. Behav. Neurosci.*, vol. 9, 2015, Accessed: Feb. 28, 2022. [Online]. Available: <https://www.frontiersin.org/article/10.3389/fnbeh.2015.00128>
- [21] K. Roelofs, "Freeze for action: neurobiological mechanisms in animal and human freezing," *Philos. Trans. R. Soc. B Biol. Sci.*, vol. 372, no. 1718, p. 20160206, Apr. 2017, doi: 10.1098/rstb.2016.0206.
- [22] T. E. Gladwin, M. M. Hashemi, V. van Ast, and K. Roelofs, "Ready and waiting: Freezing as active action preparation under threat," *Neurosci. Lett.*, vol. 619, pp. 182–188, Apr. 2016, doi: 10.1016/j.neulet.2016.03.027.
- [23] A. O. Hamm, "Fear, anxiety, and their disorders from the perspective of psychophysiology," *Psychophysiology*, vol. 57, no. 2, Feb. 2020, doi: 10.1111/psyp.13474.
- [24] E. Volchan *et al.*, "Immobility reactions under threat: A contribution to human defensive cascade and PTSD," *Neurosci. Biobehav. Rev.*, vol. 76, pp. 29–38, May 2017, doi: 10.1016/j.neubiorev.2017.01.025.
- [25] F. L. Lopes *et al.*, "Freezing reaction in panic disorder patients associated with anticipatory anxiety," *Depress. Anxiety*, vol. 26, no. 10, pp. 917–921, Oct. 2009, doi: 10.1002/da.20593.
- [26] A. F. Bastos *et al.*, "Stop or move: Defensive strategies in humans," *Behav. Brain Res.*, vol. 302, pp. 252–262, Apr. 2016, doi: 10.1016/j.bbr.2016.01.043.
- [27] D. C. Blanchard and R. J. Blanchard, "Chapter 2.4 Defensive behaviors, fear, and anxiety," in *Handbook of Behavioral Neuroscience*, vol. 17, R. J. Blanchard, D. C. Blanchard, G. Griebel, and D. Nutt, Eds. Elsevier, 2008, pp. 63–79. doi: 10.1016/S1569-7339(07)00005-7.
- [28] A. Löw, M. Weymar, and A. O. Hamm, "When Threat Is Near, Get Out of Here: Dynamics of Defensive Behavior During Freezing and Active Avoidance," *Psychol. Sci.*, vol. 26, no. 11, pp. 1706–1716, Nov. 2015, doi: 10.1177/0956797615597332.
- [29] J. Wendt, A. Löw, M. Weymar, M. Lotze, and A. O. Hamm, "Active avoidance and attentive freezing in the face of approaching threat," *NeuroImage*, vol. 158, pp. 196–204, Sep. 2017, doi: 10.1016/j.neuroimage.2017.06.054.
- [30] J. R. Jennings, M. W. van der Molen, and C. Tanase, "Preparing hearts and minds: Cardiac slowing and a cortical inhibitory network," *Psychophysiology*, vol. 46, no. 6, pp. 1170–1178, Nov. 2009, doi: 10.1111/j.1469-8986.2009.00866.x.
- [31] A. Löw, P. J. Lang, J. C. Smith, and M. M. Bradley, "Both Predator and Prey: Emotional Arousal in Threat and Reward," *Psychol. Sci.*, vol. 19, no. 9, pp. 865–873, Sep. 2008, doi: 10.1111/j.1467-9280.2008.02170.x.
- [32] P. A. Obrist, R. A. Webb, J. R. Sutterer, and J. L. Howard, "THE CARDIAC-SOMATIC RELATIONSHIP: SOME REFORMULATIONS," *Psychophysiology*, vol. 6, no. 5, pp. 569–587, Mar. 1970, doi: 10.1111/j.1469-8986.1970.tb02246.x.
- [33] P. J. Lang, R. F. Simons, M. Balaban, and R. Simons, *Attention and Orienting: Sensory and Motivational Processes*. Psychology Press, 2013.
- [34] D. J. Anderson and R. Adolphs, "A Framework for Studying Emotions across Species," *Cell*, vol. 157, no. 1, pp. 187–200, Mar. 2014, doi: 10.1016/j.cell.2014.03.003.
- [35] J. LeDoux, "Rethinking the Emotional Brain," *Neuron*, vol. 73, no. 4, pp. 653–676, Feb. 2012, doi: 10.1016/j.neuron.2012.02.004.

- [36] D. C. Blanchard, G. Griebel, R. Pobbe, and R. J. Blanchard, "Risk assessment as an evolved threat detection and analysis process," *Neurosci. Biobehav. Rev.*, vol. 35, no. 4, pp. 991–998, Mar. 2011, doi: 10.1016/j.neubiorev.2010.10.016.
- [37] R. J. Blanchard and D. C. Blanchard, "Attack and defense in rodents as ethoexperimental models for the study of emotion," *Prog. Neuropsychopharmacol. Biol. Psychiatry*, vol. 13, pp. S3–S14, Jan. 1989, doi: 10.1016/0278-5846(89)90105-X.
- [38] J. LeDoux and N. D. Daw, "Surviving threats: neural circuit and computational implications of a new taxonomy of defensive behaviour," *Nat. Rev. Neurosci.*, vol. 19, no. 5, pp. 269–282, May 2018, doi: 10.1038/nrn.2018.22.
- [39] D. Mobbs, C. C. Hagan, T. Dalgleish, B. Silston, and C. Prévost, "The ecology of human fear: survival optimization and the nervous system," *Front. Neurosci.*, vol. 9, 2015, doi: 10.3389/fnins.2015.00055.
- [40] M. S. Fanselow, "Conditional and unconditional components of post-shock freezing," *Pavlov. J. Biol. Sci.*, vol. 15, no. 4, pp. 177–182, Oct. 1980, doi: 10.1007/BF03001163.
- [41] M. S. Fanselow, "What is conditioned fear?," *Trends Neurosci.*, vol. 7, no. 12, pp. 460–462, Dec. 1984, doi: 10.1016/S0166-2236(84)80253-2.
- [42] D. M. L. Vianna and P. Carrive, "Changes in cutaneous and body temperature during and after conditioned fear to context in the rat," *Eur. J. Neurosci.*, vol. 21, no. 9, pp. 2505–2512, 2005, doi: 10.1111/j.1460-9568.2005.04073.x.
- [43] P. Walker and P. Carrive, "Role of ventrolateral periaqueductal gray neurons in the behavioral and cardiovascular responses to contextual conditioned fear and poststress recovery," *Neuroscience*, vol. 116, no. 3, pp. 897–912, Feb. 2003, doi: 10.1016/S0306-4522(02)00744-3.
- [44] P. Tovote, M. Meyer, A. Ronnenberg, S. O. Ögren, J. Spiess, and O. Stiedl, "Heart rate dynamics and behavioral responses during acute emotional challenge in corticotropin-releasing factor receptor 1-deficient and corticotropin-releasing factor-overexpressing mice," *Neuroscience*, vol. 134, no. 4, pp. 1113–1122, Jan. 2005, doi: 10.1016/j.neuroscience.2005.05.027.
- [45] I. Q. Whishaw, B. P. Gorny, and H. C. Dringenberg, "The Defensive Strategies of Foraging Rats: A Review and Synthesis," *Psychol. Rec.*, vol. 41, no. 2, pp. 185–205, Apr. 1991, doi: 10.1007/BF03395105.
- [46] B. S. Kapp, P. J. Whalen, W. F. Supple, and J. P. Pascoe, "Amygdaloid contributions to conditioned arousal and sensory information processing," in *The amygdala: Neurobiological aspects of emotion, memory, and mental dysfunction*, New York, NY, US: Wiley-Liss, 1992, pp. 229–254.
- [47] P. J. Lang and M. Davis, "Fear and anxiety: animal models and human cognitive psychophysiology," *J. Affect. Disord.*, p. 23, 2000.
- [48] T. Butler *et al.*, "Human fear-related motor neurocircuitry," *Neuroscience*, vol. 150, no. 1, pp. 1–7, Nov. 2007, doi: 10.1016/j.neuroscience.2007.09.048.
- [49] G. Griebel, D. C. Blanchard, and R. J. Blanchard, "Evidence that the Behaviors in the Mouse Defense Test Battery Relate to Different Emotional States: A Factor Analytic Study," *Physiol. Behav.*, vol. 60, no. 5, pp. 1255–1260, Nov. 1996, doi: 10.1016/S0031-9384(96)00230-2.
- [50] J. Vila *et al.*, "Cardiac defense: From attention to action☆," *Int. J. Psychophysiol.*, vol. 66, no. 3, pp. 169–182, Dec. 2007, doi: 10.1016/j.ijpsycho.2007.07.004.
- [51] A. P. Swiercz, A. V. Seligowski, J. Park, and P. J. Marvar, "Extinction of Fear Memory Attenuates Conditioned Cardiovascular Fear Reactivity," *Front. Behav. Neurosci.*, vol. 12, p. 276, Nov. 2018, doi: 10.3389/fnbeh.2018.00276.
- [52] M. M. Hashemi *et al.*, "Neural Dynamics of Shooting Decisions and the Switch from Freeze to Fight," *Sci. Rep.*, vol. 9, no. 1, p. 4240, Dec. 2019, doi: 10.1038/s41598-019-40917-8.

Appendix B

Dear Dr. Brosnan,

Thank you very much for accepting our revised manuscript for publication pending minor revisions. We are grateful for the helpful comments and the transparent review process that was instrumental in improving the article and clarifying its rationale and implications. We now addressed the few remaining points raised by the third reviewer. Please see our responses to these comments below.

With these changes, we hope we have satisfactorily provided answers to all concerns and questions. We are looking forward to seeing our paper published in *Proceedings of the Royal Society B*.

Sincerely,

Alma Merscher

Associate Editor

Comments to Author:

The two reviewers who reviewed your original submission have now reviewed this resubmission. Both are in agreement that you have carefully responded to their original feedback and both praise the revisions that you have made. I concur. I find your revised article to be much more clear and compelling. In addition to these reviewers a third (new) reviewer has also provided feedback on this version of your article. They provide some feedback on the framing of your results that I believe you should be able to address relatively easily (while they suggest that additional controls might be beneficial, I agree with them that they are not necessary.)

Thank you very much for your positive feedback on our revised manuscript. We are happy that we have successfully addressed the concerns raised by the original two reviewers and we will address the new feedback by the third reviewer below.

Referee: 2

Comments to the Author(s).

With the thorough changes by the authors, the manuscript has become more concise and the results can be evaluated more clearly. Perhaps even more important, however, the integration of the current work into a bigger defense/approach framework has - in my view - significantly improved the manuscript, which will now serve as a spark for future integrative psychophysiological research on motivated responding and emotional reactivity. All my comments have been appropriately addressed.

I want to thank the authors for the extensive responses, but even more for the intriguing manuscript, that will surely make an important contribution to the field.

We greatly appreciate your positive evaluation and are happy that we have been able to integrate the desired changes.

Referee: 1

Comments to the Author(s).

The authors did a great job addressing my questions and concerns.

Thank you very much for this positive feedback and the helpful suggestions in the first round of reviews!

Referee: 3

It seems that I have been asked to review a revised version of a manuscript, and I am painfully aware that sometimes this can result in double (or worse: completely

contradictory!) comments. I have gone through the manuscript with this in mind, and do hope that the authors forgive me any inconveniences caused as a consequence of the situation.

In general, I thought this was an interesting manuscript. The authors present the idea that "freezing"-like responses could be preserved in humans in the oculomotor system, and do indeed seem to find evidence for this - but only when a potential reward or threat was contingent on subsequent action.

I am not sure whether I agree with the authors' chosen title and framing of "centralised gaze as a threat-specific component of defensive states". This is for two main reasons, outlined below.

Signed,
Edwin Dalmaijer

We greatly appreciate your empathetic response. Thank you! We understand your concerns and revised our manuscript accordingly.

1) Centralised gaze was not threat-specific, but also appeared in reward. I appreciate that the authors show that this was to a lesser extent in horizontal directions, and I buy into their interpretation that this was a strategy by participants to look towards the potential location of stimuli. It's nicely analogous to goal and sign tracking in rodent experiments.

The authors argue that if the centralised gaze before threat was strategic, one would also see such a response in the reward experiment. Instead, they see a different strategy. This glosses over the fact that the oculomotor system responds differently to gaining something beneficial compared to risking something adverse. For example, Muhammed and colleagues (2020, <https://doi.org/10.1016/j.cortex.2018.12.001>) showed that saccadic velocity can speed up for contingent rewards, but not for contingent losses.

Obviously, being shocked is not the same as a monetary loss. However, I would argue that avoiding a potential loss is a better control for Experiment 1's avoiding an adverse stimulus. Without more controls, I'm not entirely convinced of the strong claim of centralised gaze being fully threat-specific.

One potential remedy would be to run an extra experiment with losses instead of rewards. This comes with the obvious downside of being a huge effort, especially in times of COVID. I don't think this type of time/effort investment is warranted. Instead, I would argue that the authors should stress the limitations of their control experiment, and to tone down their claims of central gaze being threat-specific.

We agree that an additional control experiment using losses instead of rewards seems relevant for further substantiating the threat-specificity of the observed oculomotor dynamics. We included this aspect as a limitation in our Discussion section and changed the title of the manuscript accordingly.

Title:

Centralized gaze as an adaptive component of defensive states in humans

Discussion, lines 417-420:

“As oculomotor responding has previously been shown to differ between gaining rewards and risking losses [1], a direct comparison comparison between (monetary) losses and shocks in future studies could be one way of further exploring the threat-specificity and defensive nature of this effect.”

2) It is somewhat unclear to me whether the gaze is a component of defensive states, or rather a by-effect of attentional processes. I realise that this sounds like a comment brought up by Reviewer 1, but I would like to elaborate on why I'm unsure even after reading the authors' responses to them.

Humans show strong attentional capture for threat. This is true when conditioned stimuli are used as task-irrelevant distractors (Mulckhuysen & Dalmaijer, 2016, <http://dx.doi.org/10.3758/s13415-015-0391-2>). In fact, oculomotor bias for threat-associated stimuli persists after successful extinction; even when self-reported expectancy and pupil dilation have reduced back to baseline (Armstrong et al., 2022, <https://doi.org/10.31234/osf.io/4ct8x>).

With the above in mind, it seems plausible participants were simply focussed on the threat-related movement they were about to make. Is this a "defensive state", or is it attention turning inwards in preparation for a movement? Or is the attentional shift a part of the "anticipatory defensive state"?

It might be due to a word limit, but the Discussion currently reads as though the authors are trying to sell their results as identifying a unique element of threat avoidance, while alternative or more nuanced explanations aren't fully explored.

We understand your concern and added this consideration to the Discussion section.

However, we still argue that an initial classification of this response as a defensive behavior is adequate, since the observed restriction of oculomotor behavior resembles an inhibition of bodily movement that is usually adduced to indicate freezing-like responses in humans.

Moreover, we did not observe such globally reduced visual exploration during the anticipation of reward, thus ruling out that it is a general phenomenon of movement preparation. However, we see your point and expanded the Discussion section to appropriately acknowledge it.

Discussion, lines 412-424:

“Importantly, the unique occurrence of globally reduced visual scanning (i.e., on both the horizontal and the vertical axis) in a threatening context, that did not merely reflect task-specific demands [38] and predicted the speed of avoidance response, highlights the adaptive and defensive nature of this oculomotor component. Whether this centralization of gaze is indeed a threat-specific component of the defensive state itself or an attentional shift as part of an anticipatory state has yet to be conclusively addressed. As oculomotor responding has previously been shown to differ between gaining rewards and risking losses [1], a direct comparison comparison between (monetary) losses and shocks in future studies could be one way of further exploring the threat-specificity and defensive nature of this effect. Importantly, similar to reductions in body sway, the currently observed decrease of oculomotor activity can be characterized as inhibition of motion, which, due to similar temporal dynamics, allows an initial classification of these responses to reflect freezing-like behavioral states that promote fast subsequent defensive actions [33], [50]. “

Discussion, lines 429-431:

“In general, centralization of gaze may best reflect oculomotor changes during the expectation of avoidable threat and future studies should explore whether these reductions in visual exploration correlate with reduced bodily movement [5].”

MINOR POINTS

Very minor point, but the authors copied the Figure 2 legend into Figure 4, but it seems they forgot to include the shock/money icons in the actual figure. (The dotted line for their onset is still visible.)

Thanks a lot for pointing this out. Figure 4 now also includes the shock/money icons.

Trial type

- Avoidable shock / achievable reward
- Inevitable shock / guaranteed reward
- No shock/reward
- (Avoidable) shock
- (Achievable) reward

Comparisons

- Avoidable shock / achievable reward vs. inevitable shock / guaranteed reward
- Avoidable shock / achievable reward vs. no shock/reward

Fig. 4. Autonomic responses during the anticipation of an inevitable, no or avoidable shock in Experiment 1 (heart rate: A, skin conductance: B, pupil width: C) or a guaranteed, no or achievable reward in Experiment 2 (heart rate: D, skin conductance: E, pupil width: F). Shaded ribbons denote standard errors of the mean. Horizontal lines at the top of each figure indicate significant differences between avoidable shock (A, B, and C) or achievable reward (D, E and F) trials and the other two trial types (after false discovery rate correction). Shading in grey denotes the phase between onset and offset of picture presentation with the offset prompting quick responses in the avoidable shock and achievable reward trials, respectively.

References

- [1] K. Muhammed, E. Dalmaijer, S. Manohar, and M. Husain, “Voluntary modulation of saccadic peak velocity associated with individual differences in motivation,” *Cortex*, vol. 122, pp. 198–212, Jan. 2020, doi: 10.1016/j.cortex.2018.12.001.
- [2] J. Anderson, H. B. Barlow, R. L. Gregory, M. F. Land, and S. Furneaux, “The knowledge base of the oculomotor system,” *Philos. Trans. R. Soc. Lond. B. Biol. Sci.*, vol. 352, no. 1358, pp. 1231–1239, Aug. 1997, doi: 10.1098/rstb.1997.0105.
- [3] A. Löw, M. Weymar, and A. O. Hamm, “When Threat Is Near, Get Out of Here: Dynamics of Defensive Behavior During Freezing and Active Avoidance,” *Psychol. Sci.*, vol. 26, no. 11, pp. 1706–1716, Nov. 2015, doi: 10.1177/0956797615597332.
- [4] T. E. Gladwin, M. M. Hashemi, V. van Ast, and K. Roelofs, “Ready and waiting: Freezing as active action preparation under threat,” *Neurosci. Lett.*, vol. 619, pp. 182–188, Apr. 2016, doi: 10.1016/j.neulet.2016.03.027.
- [5] K. Roelofs, “Freeze for action: neurobiological mechanisms in animal and human freezing,” *Philos. Trans. R. Soc. B Biol. Sci.*, vol. 372, no. 1718, p. 20160206, Apr. 2017, doi: 10.1098/rstb.2016.0206.